# Covid19/IT the digital side of Covid19: A picture from Italy with clustering and taxonomy

Vincenzo Bonnici[1], Giovanni Cicceri[2], Salvatore Distefano[2]*, Letterio Galletta[3], Marco Polignano[4], Carlo Scaffidi[2]

1 Università degli Studi di Parma, Parma, Italy, 2 Università degli Studi di Messina, Messina, Italy, 3 Scuola IMT Alti Studi Lucca, Lucca, Italy, 4 Università degli Studi di Bari Aldo Moro, Bari, Italy

☯ These authors contributed equally to this work.
* sdistefano@unime.it

**Data Availability Statement:** All files are available at https://doi.org/10.5281/zenodo.5753581 and https://github.com/marcopoli/covid19_clustering.

**Funding:** The authors received no specific funding for this work.

## Abstract

The Covid19 pandemic has significantly impacted on our lives, triggering a strong reaction resulting in vaccines, more effective diagnoses and therapies, policies to contain the pandemic outbreak, to name but a few. A significant contribution to their success comes from the computer science and information technology communities, both in support to other disciplines and as the primary driver of solutions for, e.g., diagnostics, social distancing, and contact tracing. In this work, we surveyed the Italian computer science and engineering community initiatives against the Covid19 pandemic. The 128 responses thus collected document the response of such a community during the first pandemic wave in Italy (February-May 2020), through several initiatives carried out by both single researchers and research groups able to promptly react to Covid19, even remotely. The data obtained by the survey are here reported, discussed and further investigated by Natural Language Processing techniques, to generate semantic clusters based on embedding representations of the surveyed activity descriptions. The resulting clusters have been then used to extend an existing Covid19 taxonomy with the classification of related research activities in computer science and information technology areas, summarizing this work contribution through a reproducible survey-to-taxonomy methodology.

## Introduction

This nightmare started in late 2019 in China, when the world discovered and identified the SARS-CoV2 coronavirus. In the beginning it was considered as *"just another new flu"*, but years later, at the time of this writing, it is not yet over and can be surely included among the worst pandemics ever [1], in the hopes of it will not be the worst one. This new SARS-CoV2 virus, indeed, spreads very fast neglecting geographic boundaries, worldwide. In few months, on March 2020, the world dramatically realized about Covid19, once the WHO categorized it as a pandemic, with more than a half-million people infected and nearly 30000 deaths. In late February/early March 2020, Italy was one of the first European/western country strongly

**Competing interests:** The authors have declared that no competing interests exist.

affected by the Covid19 pandemic, when most of Italians, shocked by the statistics on infections and deaths, started experiencing pandemic best practices and policies for the outbreak management such as wearing masks, careful hand-washing, social distancing, contact tracing, and lock down. Countries across the world, indeed, declared mandatory stay-at-home measures, closing schools, businesses, and public places.

Thereby, the push for the human race to survive the pandemic became the primary concern in the world and thousands of companies and even more researchers began working on solutions, tests, treatments, and vaccines. In this context, the computer science, engineering and technology community played an important role, providing mechanisms and tools that supported, directly or indirectly, the fight against Covid19. Medical imaging techniques, contact tracing apps, smart working, distance learning and similar solutions are just some examples of IT-based Covid19 tools. Also virology, epidemiology as well as several other (medical and non-medical) disciplines adopted computer-based solutions in approaching Covid19 related problems. For example, Covid19 vaccines and any other medical treatments exploited information technologies (IT) in, e.g., identifying potential medication molecules, simulating and assessing their effects, probing and managing clinical data from experiments, mining such data.

Motivated by the local pandemic outbreak, the Italian computer science and engineering research community pioneered this trend, among the first in promptly reacting to the Covid19 pandemic with several multidisciplinary contributions. To document this effort, in May 2020, IT-based activities against Covid19 in Italy have been surveyed by a task force established in CINI (National Interuniversity Consortium for Informatics—https://www.consorzio-cini.it/index.php/en/), a public consortium aggregating almost 50 universities and research centers active in computer science and engineering across Italy. The main goal of this paper is to document and report on such activities, by sharing and discussing the survey results, and then analyzing and further elaborating them. More specifically, the responses obtained by the survey have been first collected into a dataset composed of tuple of textual information. These (raw) data have been then analyzed to show some descriptive statistic with related considerations and thus, once filtered, cleaned, and aggregated, further processed by exploiting NLP methodologies, techniques and tools to revise the initial classification into a taxonomy of Covid19 IT-based research activities. The resulting taxonomy partly confirmed the initial classification adopted in the survey, reducing the original categories and slightly revising them.

## Relevance of the topic

At the time of writing, Covid19 is still a pandemic with millions of cases per day. This paper investigates on it from a technological perspective, aiming to identify new solutions and ways to exploit information technologies which may be effective in contrast to Covid19 and other pandemics. IT mechanisms and tools, indeed, have been adopted in different contexts and applications, to either support other domains (virology, epidemiology, telemedicine, biology, pharmacy, chemistry, psychology, decision makers) or directly as ready-made solutions (smart working, distance learning, contact tracing). This fact highlights how IT is of strategically importance in the fight against pandemics, and how to identify, categorize, organize and rationalize such solutions can be helpful for future efforts, against the Covid19 pandemic and those to come.

## Novelty

In general terms, the novelty of this paper is the methodology adopted in conducting the research, starting from a survey of Covid19-related IT initiatives in Italy and identifying as

main target the specification of a taxonomy from the thus collected textual data. This survey-to-taxonomy methodology is described in a subsection below, implemented by a workflow that can be generalized to similar efforts. In more specific terms, the novelty of this paper is 4-fold:

i). a case study, a deep investigation on Covid19 activities of the Italian research community on computer science, engineering, and IT, with results and statistics;

ii). the dataset obtained from the survey, released to the research community for further investigation (see https://doi.org/10.5281/zenodo.5753581);

iii). the classification of such activities by adopting natural language processing and clustering techniques, comparing different models to identify 12 clusters of activities as a result after in depth quality assessment analysis; and

iv). the Covid19/IT ontology extending the one of [2] with Covid19 IT-related research categories.

## Research impact and policy implications

This work aims at impacting on the research, business, social and political communities in different ways, with several potential applications and exploitation. In general, this paper research artifacts, with specific regard to the survey dataset and the ontology, as well as the analysis of the survey results and related considerations, can inform all such communities and the public opinion on the corresponding activities, then even supporting them in decision making. More specifically, from a research perspective, the above contribution can be the basis for new researches, fostering cross-fertilization for multi-/inter-/trans-disciplinary collaborations, starting from the dataset and the ontology publicly provided in this paper. The main goal of this paper is, indeed, to organize, rationalize and classify the knowledge, areas, disciplines and application contexts of IT-related efforts against Covid19 and pandemics into a taxonomy. From a business perspective, this work could be a reference for knowing the state of the art on information technology applications against Covid19, thus a starting point for new products and services avoiding to "reinvent the wheel", while exploiting the business agility of smart information technologies, which can also allow to be *resilient* to events like pandemics in the future [3]. From a social perspective, this paper can bring evidence on how technological is the fighting against Covid19 and pandemics, informing and somehow encouraging people to have a better outlook on future pandemics. From a political viewpoint, this paper aims at becoming a reference to support decisions, providing the IT baseline and some related guidelines on how to apply such technologies to future epidemics and pandemics, highlighting the importance of IT in such context.

## Methodology, theoretical framework and limitations of the study

The methodology adopted in this work is represented by the workflow of Fig 1 describing its research process organized into 4 phases: 1) the *Surveying* phase, in which the survey has been designed, implemented (*Survey Design and Implementation*) and opened to the respondents (*Publication and Contribution*) collecting their responses (*Data Ingestion and Collection*); 2) the *Assessment* stage, where the collected data have been analyzed (*Descriptive (Raw) Data Analysis*), issuing a full report with statistics and descriptions on all activities (*Reporting*); 3) the *Preprocessing and Clustering* stage where such data have been filtered and cleaned to improve the their quality for further analysis (*Data Filtering and Cleansing*) aiming to revise

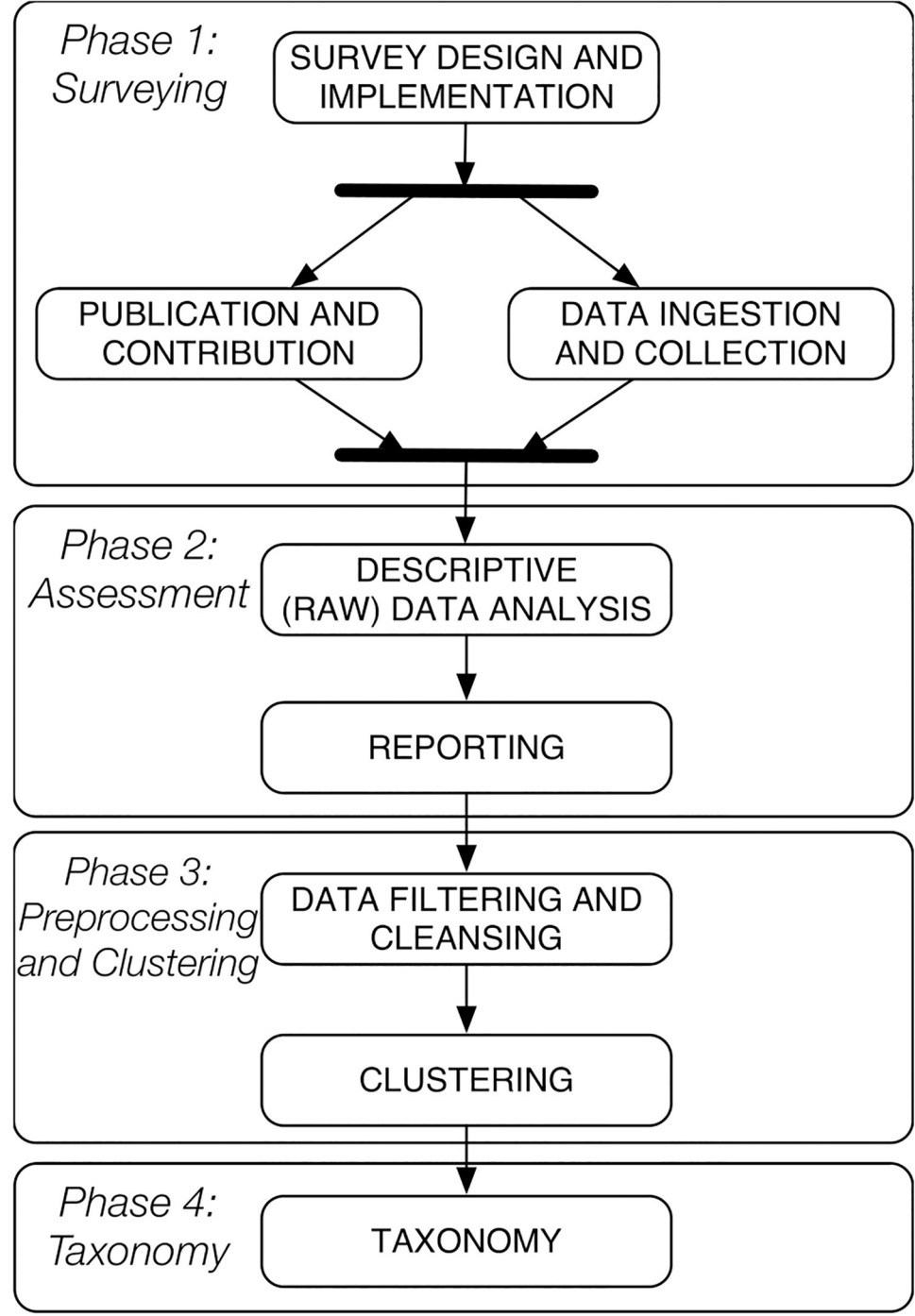

**Fig 1. Workflow of this research.**

the original (manual) classification of activities into a set of clusters through NLP techniques and tools (*Clustering*); and finally in the *Taxonomy* phase 4), the new classification and the categories thus identified have been included into a Covid19 ontology extending [2] with new concepts from the IT research on Covid19. As discussed above, this methodology is general

enough to be applied to similar efforts, even in different contexts, thus considering the methodology itself as a relevant contribution of this paper.

In conducting the survey as well as in analyzing the obtained results we based on similar initiatives [2, 4–6], mainly adoting natural language processing solutions such as "tf-idf", vectorization (using both word2vec [7] and BERT [8]), and K-means clustering [9], comparing different alternatives and configurations. The final ontological step has been then performed starting from an existing Covid19 ontology [2], then extended towards the IT domain with the new concepts and classes identified by the clustering step.

Limitations and threats to validity of the present work can be mainly summarized into three points: i) the size of the dataset (128 activities/entry surveyed), ii) the data quality, and iii) the Italian language used in the responses. A larger dataset with a more detailed description of the initiatives allow to improved the overall accuracy of the clustering algorithm and of its results as well. In addition, the Italian language used in textual descriptions does not allow to reuse the datasets in more International context that, since NLP algorithms achieve better performance on English texts, would also be a way to further improve the overall results.

## Contribution to the literature

Based on the 4 main contributions above identified as novelty for this paper, the literature enhancements can be specified as follows

i). With regard to the survey, it collects data on Italian IT initiatives against Covid19. Despite focusing on IT and computer science, this enlarges previous work scopes, summarized in the next section, which are mainly restricted to a specific area of a scientific discipline (partly covering IT), with the main goal of sharing resources and information on the corresponding community.

ii). dataset on Italian IT initiatives against Covid19 here released is, to the best of our knowledge, the first focusing on IT and computer science from a general perspective. Other datasets [4, 10–14] focus on specific Covid19 topics and are mainly designed for machine learning algorithms.

iii). Word embedding, NLP and clustering techniques have been applied to the textual descriptions of the surveyed initiatives to categorize them into semantically homogeneous groups. To the best of our knowledge, such techniques have been used in different domains, such as law [15] and economics [16], but this is one of the first attempts to clustering initiatives against Covid19 from textual descriptions.

iv). An existing Covid19 ontology [2] has been extended by enriching it with new concepts from IT and computer science into the Covid19/IT one here proposed, classifying such Covid19-related IT activities into specific categories.

## Outline

The reminder of the paper, after a review of the state of the art on similar initiatives, surveys and taxonomies related to Covid19 and IT, is therefore organized following the workflow phases of Fig 1. Hence, the survey activities, including the definition of the dataset features, are first described (phase 1) and then, from the responses thus collected, some statistics and considerations on the survey results are provided (phase 2). Preprocessing and clustering steps (phase 3) are described afterwards as well as the Covid19/IT ontology obtained from them (phase 4). Some final remarks and considerations then close the paper.

## Related work

Several initiatives in response to the Covid19 pandemic, directly or indirectly involving professionals and scientists, have been proposed since the pandemic outbreak in late 2019-early 2020. In this section, we provide a brief description of the most relevant and innovative ones. First, working groups, coalitions and task forces established in academic and industrial contexts against Covid19, are reported. Then, IT solutions against Covid19 are considered and finally datasets, taxonomies, and ontologies concerning different pandemic aspects are reviewed.

### Groups, coalitions and task forces against Covid19

Several working groups and task forces have been created so far to share ideas, resources, and solutions to contrast the pandemic. Some among the most relevant ones are summarised in Table 1. results have not yet been published because still under review or not complete. The first initiative considered is pursued by the *Confederation of Laboratories for Artificial Intelligence Research in Europe* (CLAIRE) [17] task force [18]. It promotes and coordinates research projects aimed at creating tools and techniques based on artificial intelligence to support the fight against Covid19. The CLAIRE task force identifies seven working groups addressing different aspects of Covid19-related topics ranging from data analysis for epidemiological models to the study of clinical, biological, and molecular data, up to the adoption of robots in Covid19 clinics. Several groups of the task force conducted survey activities, for example, the group on Medical Imaging, whose results have not yet been published because still under review or not complete.

A similar initiative is the *Covid19 Open AI Consortium* [19], joining academic and industrial partners with data mining and AI background to fight Covid19. Specifically, this consortium aims at fostering collaboration on joint research projects, supporting the development of effective treatments against the virus, and establishing a reference platform to exchange results and findings. The consortium works on several areas, some of which focus on the classification and prediction of the immune response to the virus and on its effects on the organs.

In the context of medical imaging, the *Imaging Covid19 AI Initiative* [20] involves several European research centers with the aim of enhancing tomography techniques for Covid19 diagnostic through artificial intelligence. This initiative aims to define and train machine learning models with (anonymized) tomographic images to improve and automate the diagnosis of Covid19, even estimating its severity and impact on patient organs. Although not exhaustive, a survey on techniques and studies on this topic is provided by their website.

The *Covid19 High Performance Computing (HPC) Consortium* [21] collects both industrial and academic partners to share their high-performance computing resources, facilities, services, platforms, and skills to support Covid19 research projects. Similarly, *HPC vs Virus* [22] is an European initiative supported by the *Partnership for Advanced Computing in Europe* (PRACE) [23] providing computational resources for processor-intensive algorithms and models. PRACE is a non-profit association that includes 26 European partners, opening a fast track call for proposals and projects against Covid19.

In Italy, the *Istituto di Robotica e Macchine Intelligenti* (Italian Institute of Robotics and Intelligent Machines) [24] carried out a survey on Covid19 and robotics, then sharing the results by a database available online. Surveyed initiatives mainly focus on the development and testing of robotic solutions in response to Covid19. This survey categorizes such initiatives by the application context, the geographic scope, the TRL, the status of development, and the type (research project or product).

**Table 1. Coalitions established to fight Covid19.**

| Name | Homepage | Promoters | Research Topics | Scope | Goals | Report |
|---|---|---|---|---|---|---|
| TFCovid19AI | https://covid19.claire-ai.org/ | CLAIRE | Articial Intelligence | European | Tools to support the fight against the virus and the illness | Under review |
| Covid19 Mass Spectrometry Coalition | https://covid19-msc.org/ | Different International Labs | Mass Spectrometry | International | • Sharing of methods, protocols, and data among members<br>• Mapping of viral antigens in blood and other fluids to support serological testing<br>• Support with mass spectrometry the development of vaccines and therapies | None |
| Covid19 Host Genetics Initiative | https://www.covid19hg.org/ | Different International Labs | Genetics | International | • Creation of a platform for sharing research projects and resources<br>• Organise analytical activities among various studies to determine genetic determinants of the virus<br>• Support the sharing of scientific results obtained | Only an overview |
| Covid19 Open AI Consortium | https://owkin.com/covid-19-open-ai-consortium/ | OWKIN | • Artificial Intelligence<br>• Medical research | International | • Promote and support collaborative research on Covid-19<br>• Accelerate clinical development of treatments<br>• Support the sharing of results with the scientific community | None |
| Imaging Covid19 AI initiative | https://imagingcovid19ai.eu/ | • European Society of Medical Imaging Informatics<br>• Robovision<br>• Quibim | • Artificial Intelligence<br>• Image Processing | European | Train AI algorithms with data from several European laboratories and hospitals for automatic diagnosis of Covid19 | Only an overview |
| Covid19 High Performance Computing (HPC) Consortium | https://covid19-hpc-consortium.org/ | • IT big companies<br>• Different American Universities | High performance computing | USA | Provide computational resources, services, and expertise to support scientific research against Covid19 | None |
| HPC vs Virus | https://prace-ri.eu/hpc-access/hpcvsvirus/ | PRACE Consortium | High performance computing | European | Provide computational resources, services, and expertise to support scientific research against Covid19 | None |
| Covid-19 Pilot Projects | https://i-rim.it/it/progetti-pilota | Italian Institute of Robotics and Intelligent Machines | Robotics | Italy | Census of technology demonstrators that can be used in the clinical setting in response to Covid19 | https://i-rim.it/it/progetti-pilota |

The *Covid19 Mass Spectrometry Coalition* [25] brings together several mass spectrometry laboratories worldwide with the aim of sharing data, and protocols, and defining tools and methodologies to support the development of serological tests and vaccine development. Several resources (ideas, projects, datasets, etc.) on Covid19-related mass spectrometry initiatives are shared on their website, providing an overview of the work carried out by the different partners of the coalition.

The *Covid19 Host Genetics Initiative* [26] is a coalition that brings together several members of the international geneticist community to share and analyze data on biological aspects of the SARS-CoV-2 virus. The main goals of the initiative are i) to provide a platform for sharing resources to support genetic researches on the virus; ii) to foster collaborations and joint studies among the involved research centers on the identification of genetic markers and determinants for Covid19 susceptibility and severity; and iii) to disseminate the results obtained by such studies. The members of this coalition periodically release updated genomic and

statistical data on the virus and Covid19 patients through the coalition web page. Although they do not conduct a survey on Covid19-related genetic initiatives, the information published on the web page provides a quite comprehensive overview of the genetic research activities on Covid19.

All the groups, task forces, and coalitions above surveyed, are focused on a specific topic within a scientific discipline, and mainly aim at sharing resource and information to contrast the pandemic. The Covid19/IT task force, instead, focused on a wider scope, IT and computer science, with the goal of providing a quite representative list of initiatives against Covid19 on the Italian territory, in all topics related to computer science, engineering and technologies.

## IT solutions against Covid19

Recently, a large number of survey articles discussing IT solutions to address and mitigate the pandemic have been published. Most of them are based on Big Data and Artificial Intelligence. Jianguo Chen et al. [27], Quoc-Viet Pham et al. [28] reports on different approaches based on these technologies in the Covid19 context. Dinh Nguyen et al. [29] first provide an extensive survey on approaches that use blockchain and AI technologies against Covid19. Then, they introduce a conceptual architecture that integrates these two technologies for fighting the virus. In particular, their idea is to use blockchain to share medical data, ensuring AI models can detect Covid19 symptoms and support treatments and drug manufacturing starting from such blockchain data. Ting et al. [5] discuss on the application of IoT, big data, AI, and block-chain technologies to support public-healthcare system strategies for Covid19 outbreak containment, with a focus on monitoring, surveillance, detection, and prevention of viruses, as well as on mitigating the impact of the virus to healthcare. Chamola et al. [30] briefly survey Covid19 health implications, highlighting its impact on the global economy, and discussing how technologies such as IoT, Unmanned Aerial Vehicles (UAVs), blockchain, AI, and 5G can help mitigating the pandemic impact. One of the IT solution that attracted attention in the Covid19 outbreak containment is contact tracing. The paper by Jinfeng Li and Xinyi Guo [6] and the one by Qiang Tang [31] surveyed the leading contact tracing solutions proposed in the last years, focusing on user privacy issues. Similarly, Musa Ndiaye et al. [32] survey IoT-based solution to address different issues of the pandemic, mainly focusing on contact tracing, arguing on how the pandemic may influence the development of new architectures for the IoT.

In this regard, the goal of the present paper is to provide a survey, taking a snapshot of intiatives against Covid19 in Italy, including all computer science, engineering and technology topics. The papers above discussed restrict the scope to a specific computer science and engineering area.

## Covid19 datasets, taxonomies and ontologies

Many Covid19 datasets have been published so far with the aim of sharing knowledge on pandemic-related topics. Some of such datasets are designed to be mainly used by humans, others by machine learning algorithms and some can be used by both.

The Virus Outbreak Data Network (VODAN) [33] is a project of the GO FAIR Foundation that aims at making the SARS CoV-2 virus data Findable, Accessible, Interoperable and thus Reusable (FAIR) by both humans and machines. The VODAN idea is to follow the Clinical Research Form and WHO standards to represent Covid19 data. The Lean European Open Survey on SARS-CoV-2 (LEOSS) [34] is an European federated cohorts and research center, aiming to get in-depth knowledge about the epidemiology and clinical course of patients infected by SARS-CoV-2. Their main idea is to store anonymized patient data to identify Covid19 predictors.

The European COVID-19 Data Platform [35] is the European portal for the collection of Covid19-related data and scientific literature, providing state of the art solutions against Covid19. Such data include sequences, structures, expression data, compound screens, biochemistry and scientific publications. COVID-19 Data Exchange [36] is a platform, supported by many companies, allowing its participants to securely store and share Covid19 data. Specifically, this platform include statistical data, research data, anonymized raw clinical data, test results, equipment configuration and inventory data, social and sentiment analysis data, and similar. It allows users to have full control of the data they share and to keep track of who and how is operating on such data.

The CLAIRE Covid19 task force provides a list of different public available datasets and databases related to Covid19 on its web page [4]. For each dataset, metadata about the name, a short description, topics, and the publisher are provided. Many of these datasets concern biological and epidemiological data, and X-ray or CT images. Junaid Shuja et al. [37], conducted a survey and proposed a taxonomy on Covid19 open-source datasets released by the scientific community to train machine learning algorithms. Similarly, several studies [10–14] released machine learning compliant datasets on atmospheric pollutants (NO2, PM2.5, PM10 and CO2) and geolocated information about Covid19 outbreak in different parts of the world, e.g., US, India, China and France, to investigate possible correlations.

Differently from above, the dataset collected and published in this paper specifically concerns initiatives proposed by the Italian IT scientific community to contrast the Covid19. Its goal is to take a snapshot on how the Italian IT research community reacted to Covid19 in the first phases of the outbreak. Therefore, it does not provide any clinical data on Covid19, it is not focused on the problem but mostly on solutions. To such a purpose, some interesting attempts on defining taxonomies and ontologies on Covid19 related information and solutions, from different perspectives, have been proposed in the literature. Most of them mainly refer to the *medical-biological-clinical* area. For example, Helmy et al. [38] present a taxonomy of coronaviruses based on a phylogenetic analysis and genome structure identifying four families. For each family they report different peculiar features like the species, the genus and the reservoir host. Sargsyan et al. [39] propose an ontology covering molecular and cellular entities in virus-host-interactions, in the virus life cycle, as well as a wide spectrum of medical and epidemiological concepts linked to Covid19. The resulting ontology is publicly available online and suited for drug re-purposing towards Covid19 therapeutic development. Babock et al. [40] extend the Infectious Disease Ontology (IDO), a suite of inter-operable modules of ontology covering all aspects of the infectious disease domain, with new ontologies that are pathogen-specific to Covid19. Their goal is to allow data on novel diseases to be easily compared, along multiple dimensions, with data represented by existing diseases of such ontologies. Alag [41] proposes an ontology-based pipeline that mines ClinicalTrials.gov, a database of globally-conducted clinical trials maintained by the United States National Library of Medicine. The goal is to extract Covid19 related clinical trials, summarize the results in reports, and make the metadata publicly available via Application Programming Interfaces. The reports are automatically generated based on the Subject Heading terms and the Human Phenotype Ontology.

Outside the medical area, several papers propose taxonomies mainly on solutions to Covid19-related problems to support government and policymakers in the management of the outbreak response and the public healthcare. Taber et al. [42] develop a taxonomy of public health guidance related to the ongoing Covid19 pandemic across multiple organizations in the US. This taxonomy is built using a semi-automated method consisting of Web crawling and a streamlined manual analysis of the content. Zarghami [43] uses a cluster analysis to develop a taxonomy of vulnerabilities in the economic sector based on industry-specific vulnerability indicators. The goal of this taxonomy is to support public and private managers and

administrators to make decision and to establish businesses more resilient to epidemics and pandemics.

Referring to IT and computer science, Gasser et al. [44] propose a taxonomy of Covid19-related solutions adopted by public healthcare systems and governments such as proximity and contact tracing, symptom monitoring and diagnostics, quarantine control. The goal of this taxonomy is to help scientists and policy makers to understand technological and ethical implications of such applications. Similarly, Almalki and Giannicchi [45] review different app stores to analyze Covid19-related apps, categorizing them in a taxonomy based on their key technical features, goals and domains. Ahmad et al. [46] review the machine learning models used to predict the number of Covid19 cases, by a detailed analysis and a taxonomy that identifies four broad machine learning methodologies, providing hints to improve the performance of such models. Adans-Dester, together with and more than 60 specialists on wireless and mobile technologies (*mHealth*), present a taxonomy of mHealth solutions successfully applied against Covid19 [47]. They argued that mHealth technologies can be quite effective in monitoring Covid19 symptom escalation of infected people, facilitating screening and diagnosis, carrying out early intervention, and preventing exposure. Hakak et al. [48] identify Covid19-related cyberthreats and propose a taxonomy of attacks related to Covid19 and their effects on security goals, discussing on potential mitigation strategies for the identified threats. Dutta and DeBellis [49] propose CODO, an ontology for collecting and analyzing Covid19 data, providing a knowledge graph to facilitate the semantic representation and the automatic processing of such data.

It is possible to consider a taxonomy as a hierarchical classification in which semantic concepts of a well-specific domain are organized into homogeneous groups. Typically, the starting point for the taxonomy construction is a text corpus that accurately characterizes a specific domain. Robin et al. [15] describes a methodology to automatically generate a taxonomy of legal concepts, and apply this methodology on a corpus consisting of statutory instruments for the UK, Wales, Scotland, and Northern Ireland laws. Bai et al. [16] first propose a method to construct business taxonomies automatically from corporate reports. Their method extracts concept-level terms from documents, computes the similarities between different terms, and recursively cluster on the basis of their similarity. Zhang et al. [50] propose TaxoGen, an unsupervised method for constructing topic taxonomies. Calka and Bielecka [51] present a case study of constructing a taxonomy of real estate properties via agglomerate clustering and the K-means method [9]. McCrae et at. [52] report on their experiments exploiting the tool Saffron to extract a taxonomy from a text corpus in the context of financial enterprise, discussing on how the extract knowledge graph can be used. Treeratpituk et al. [53] propose a graph-based approach for constructing concept hierarchy from a text corpus.

In this context, the present paper started from the Covid19 survey response textual dataset and applied word embedding and clustering techniques to the textual descriptions to categorize the initiative into semantically homogeneous groups. From the resulting groups, a taxonomy is specified by observing the characteristics of the initiatives falling into each cluster and to extend an existing Covid19 ontology by Oyelade et al. [2] with new concepts for classifying IT related scientific efforts.

## The survey

One of the main contribution of this paper is the survey of IT-related Covid19 activities carried out in Italy during the first pandemic wave in 2020. The results obtained from the survey have been collected into a public dataset then used to specify a taxonomy through NLP processing

and clustering. In this section, the survey design, implementation and contribution steps, as well as the resulting dataset, are described.

## Design and implementation

The survey of Italian initiatives in computer science, engineering and technology against Covid19, Covid19/IT, has been conceived aiming to:

i). identify the research activities and initiatives of the Italian IT community against Covid19;

ii). provide a reference document on Covid19 IT solutions to anyone interested in the topic, in the form of a public report to be consulted according to specific needs;

iii). establish a community on Covid19/IT, fostering collaborations for new joint ideas, projects and proposals on Covid19;

iv). provide guidelines and best practices to practitioners, managers, decision makers, and professionals to support their decisions and accelerate the development processes of new solutions by reusing existing ones;

v). make aware a large audience about the importance of computer science and technology in the fight against pandemics, even after and beyond Covid19.

The survey has been implemented as a web questionnaire, i.e., an online form (https://shorturl.at/mnpGM) to be filled by the participants. The latter have been enrolled on a voluntary basis, through a dissemination campaign mainly exploiting emails and mailing lists (CINI, GRIN—GRuppo di INformatica, GII—Gruppo di Ingegneria Informatica, CNR—Consiglio Nazionale delle Ricerche, CINECA—Consorzio Interuniversitario dell'Italia Nord Est per il Calcolo Automatico—and several others).

To define the survey questions, a state of the art analysis, similar to the one reported in the previous section, has been first performed during the design. Despite not so many results were still available in April-May 2020, some of these [2, 4–6] were helpful to identify the main categories of application contexts, thus adopted as baseline for the initiative classification implemented in the report and further developed in the following. Thereby, the survey questions included a brief description of the proposed activity, the title, the contact person, a web link containing additional resources and the (8) parameters reported in Table 2, identifying common characteristics of the activities based on, as discussed above, similar initiatives, works and official taxonomies.

Despite mainly self-explanatory, some application context parameters (row id. 5 in Table 2) need clarification: 5.1) Virology and epidemiology refers to the application of computer-based solution to virology and epidemiology (e.g. models, drugs discovery); 5.2) Digital events include sports, concerts, art and public events; 5.4) E-government includes any public administration or government Covid19-related digital app or initiative; 5.6) Medical devices refer to personal devices and equipment adopted in medical contexts; 5.7) Medical imaging includes the approaches and models based on pattern detection and recognition in images using artificial intelligence or signal processing; 5.10) Prognostics and diagnostics refers to the adoption of IT and computer-based solutions to support Covid19 diagnostics and prognostics; and 5.12) Telemedicine includes solutions to support the remote interaction between doctors/caregivers and patients.

Similarly, some of the 17 scientific domain categories (row id. 6 in Table 2) need further details: 6.7) Bioinformatics encompasses the analysis of all the *omics*; 6.10) Information and

**Table 2. Parameters required to the participants of the Covid19/IT survey.**

| | | | | | | | | |
|---|---|---|---|---|---|---|---|---|
| *1. Type of initiative* | | | | | | | | |
| 1.1 Tool/Project | 1.2 Prototype | 1.3 Scientific publication | 1.4 Laboratory | 1.5 Research activity / scientific consultancy | 1.6 Dataset | | | |
| *2. Novelty: original work or reworking of existing ones* | | | | | | | | |
| 2.1 Ad hoc initiative | 2.2 Reuse and adaptation | 2.3 Ad-hoc initiative with reuse and adaptation | | | | | | |
| *3. Geographic scope of the initiative* | | | | | | | | |
| 3.1 Local (city-province) | 3.2 Regional | 3.3 National | 3.4 EU | 3.5 International | | | | |
| *4. Technology Readiness Level (1-9): maturity level of activity (1 down)* | | | | | | | | |
| *5. Application context in which the initiative is developed* | | | | | | | | |
| 5.1 Virology and epidemiology | 5.2 Digital events | 5.3 Distance learning | 5.4 E-government | 5.5 Fake news | 5.6 Medical devices | 5.7 Medical imaging | 5.8 Remote assistive technology | 5.9 Smart working |
| 5.10 Prognostics and diagnostics | 5.11 Economics | 5.12 Social services | 5.13 Scientific research services | 5.14 Smart services | 5.15 Social distancing | 5.16 Telemedicine | 5.17 Thermal screening | |
| *6. Scientific domain* | | | | | | | | |
| 6.1 Assistive technology | 6.2 Cyber physical systems | 6.3 Cyber security and privacy | 6.4 Data management systems | 6.5 Medical informatics | 6.6 Education | 6.7 Bioinformatics | 6.8 Human-computer interaction | 6.9 Software engineering |
| 6.10 Information and society | 6.11 Artificial intelligence | 6.12 Modelling and simulation | 6.13 Robotics | 6.14 Medical and life sciences | 6.15 Network services | 6.16 Sensors and actuators | 6.17 Smart cities | |
| *7. Access: describes the use-access mode of the products resulting from the initiative* | | | | | | | | |
| 7.1 Open | 7.2 Payment | | | | | | | |
| *8. Status: describes the status of the proposal, mainly in the 2 levels* | | | | | | | | |
| 8.1 Incomplete | 8.2 Finished/ready | | | | | | | |

society copes with issues related to the digital divide, digital economy, network neutrality, ethical, legal, social and epistemological aspects of informatics; 6.12) Modelling and simulation includes methods for investigating and predicting epidemics, pandemics, outbreaks and similar problems related to Covid19; 6.14) Medical and life sciences category includes formal and algorithmic methods applied to the medical domain.

## Data collection and dataset

The Covid19/IT survey has been launched on May 12th 2020 and lasted about one week. Participants have been asked to provide all the info described above. Thus, from May 12th to May 20th, 2020, 128 responses were collected from all over the country, mainly involving universities and research centers. Individuals and companies also participated, either directly, as drivers of the initiatives, or indirectly, in collaboration with scientific partners. Once the data from the responses provided by survey participants were collected, the study phase began, leading to a report summarized in this paper.

To share such results we decided to include the survey responses into a dataset, making it publicly available in the Zenodo repository (at link https://doi.org/10.5281/zenodo.5753581). A formal definition of the records contained in such a Covid19/IT dataset is expressed in

terms of the 14-tuple defined by Eq (1)

$$d^i = \langle (Ts, C, I, N, T, TRL, K, D, O, G, AC, SD, UA, S) \rangle. \tag{1}$$

where $d^i$ represents the datapoint associated with the $i$th initiative and

- $Ts$ is the *timestamp*, i.e. a DateTime object set to the submission date and time instant;

- $C$ is a textual object representing the *city* of the respondent;

- $I$ is the *type of initiative*, a textual information represented by 6 categories: *Tool/Project, Prototype, Scientific Publication, Thematic Laboratory, Research Activity / Scientific consultancy, and DataSet*;

- $N$ represents the *novelty*, a textual label represented by 3 categories: *Ad hoc initiative*, *Reuse and adaptation*, and *Ad-hoc initiative with reuse and adaptation*;

- $T$ is a textual object containing the *title* of the initiative;

- $TRL$ is the *Technology Readiness Level* (TRL), an integer number representing the maturity level of the project (1 low—9 high);

- $K$ indicates the *keywords*, a set of a comma-separated list of keywords;

- $D$ reports the textual *description* of the initiative;

- $O$ reports the initiative *objectives* as text;

- $G$ is the *Geographic Scope* of the initiative, a textual object defined as *Local (City-Province)*, *Regional*, *National*, *European* and *International* relevance;

- $AC$ represents the *Application Context* and

- $SD$ represents the *Scientific Domain*, both described by text and represented by 17 categories each (reported in Table 2) encoded by real numbers indicating their relevance level (*0:none, 0.2:low, 0.5:medium, 1:high*);

- $UA$ refers to the use-access mode (Open or Payment);

- $S$ is related to the status of the proposal (Incomplete or Finished/Ready).

The dataset $D$ has been obtained by merging all the datapoint tuples $d^i$ containing all the data provided by the survey participants. A the end of the procedure, a dataset with 128 entries has been created and shared to the community (at link https://doi.org/10.5281/zenodo.5753581).

## Descriptive data analysis

Once the responses from the survey participants have been collected, the dataset thus obtained has been analyzed by means of descriptive statistics and the results of this analysis are discussed in the following.

### Territorial statistics

We first performed a territorial investigation based on the type, the TRL and the geographic scope of the 128 surveyed activities, which may belong to more than one of the 6 categories of type above identified. A great part of such initiatives (about 39.1%) are tools and projects implementing platforms to support the public healthcare system, as it is shown in Fig 2. 23.4% of activities are research or scientific consulting activities, 16.4% of the initiatives focus on

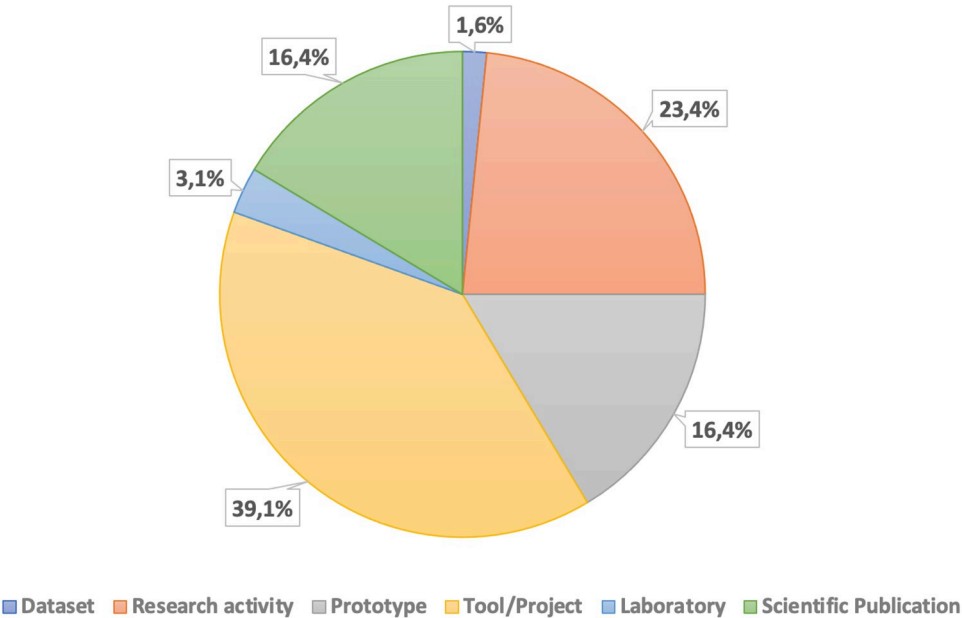

**Fig 2. Statistics on the types of the surveyed initiatives.**

application prototyping and scientific publications, 3.1% aim at establishing laboratories, while only 1.6% of the activities include a dataset as an outcome.

A similar analysis has been performed on the TRL maturity of surveyed initiatives. Each initiative is associated with only one TRL level ranging from 1 to 9, where 1 is the lowest value and represents basic research, while 9 characterizes applied research and technology adopted in commercial businesses and products. Statistics on the TRL of the surveyed initiatives are reported in Fig 3. Most of the initiatives have a medium-high TRL between 5 and 7, thus ready

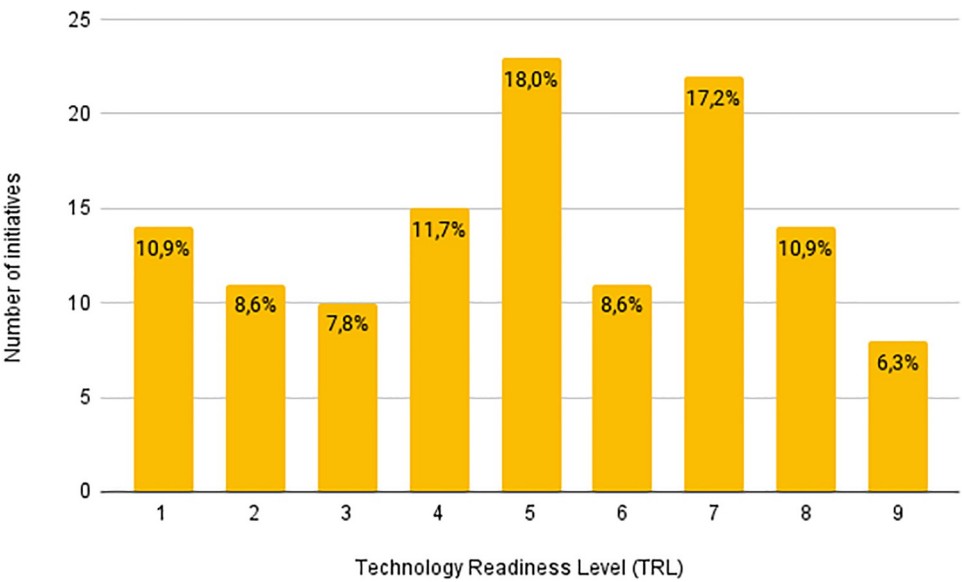

**Fig 3. Number of surveyed initiatives for each TRL level.**

and able to promptly react to the pandemic. The TRL average value of all the surveyed initiatives, also including more theoretical ones, is 5.03.

Considering the geographic scope of the surveyed initiatives, different levels ranging from local (city of province), to regional, national, European and international relevance have been identified, allowing to select multiple option. Descriptive statistics show that the surveyed activities are mainly international (32.2%), then national (24.6%), regional (16.9%), European (14.2%), and local (12%). For what concerns the correlation between scopes, all initiatives have high scale-up vocations, as shown in Fig 4 reporting the scope co-occurrences. More specifically, local activities scale up to regional and national initiatives as well as regional ones towards mainly national and European, and national to European.

Fig 5 shows a descriptive map of the geolocalized surveyed activities. The maps shows a uniform distribution of the activities all along the national territory, except for Sardinia that is the only Italian region without initiatives. There is a prevalence of initiatives in the central-northern part of the country that somehow partially reflects the distribution of universities and research centers in Italy. However, it should be taken into account that the pandemic started in the north of Italy in early 2020, and for several days the southern part was slightly affected by it, as shown by Fig 5. Thus, this descriptive analysis mainly highlights a high cause-effect correlation between the pandemic outbreak and the reported activities.

## Scientific domain statistics

As discussed above, each activity has been characterized by at least one (or more) scientific domains (among the 17 above specified), ranking them according to their relevance (*high*, *medium*, *low* or *no*) as shown in Fig 6. Fig 7 reports the statistics on the number of activities in

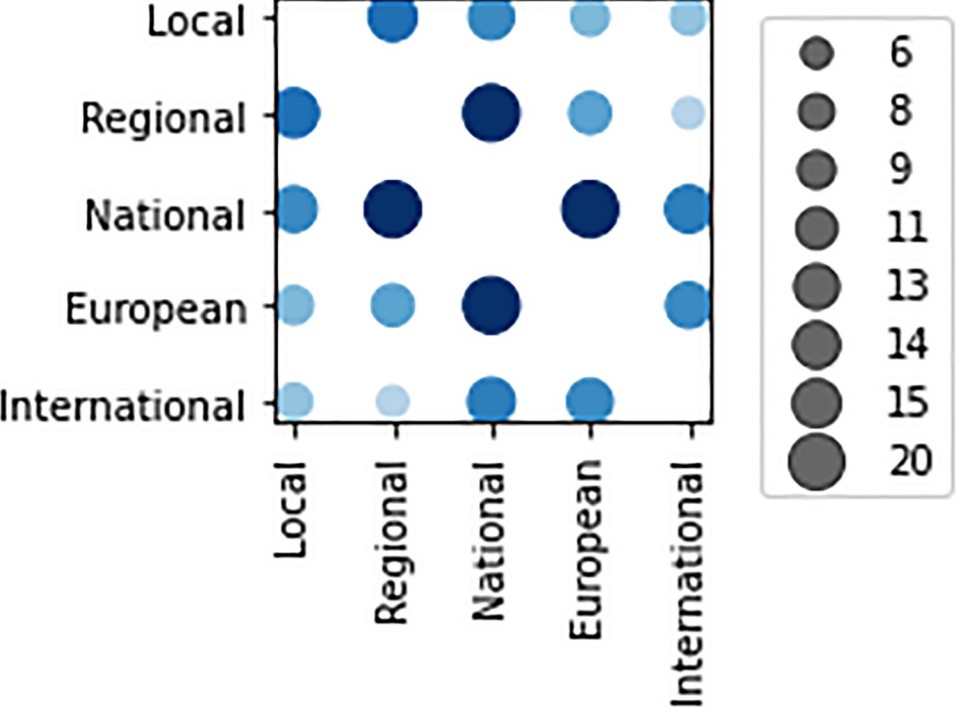

**Fig 4. Co-occurrence between levels of geographic scope.**

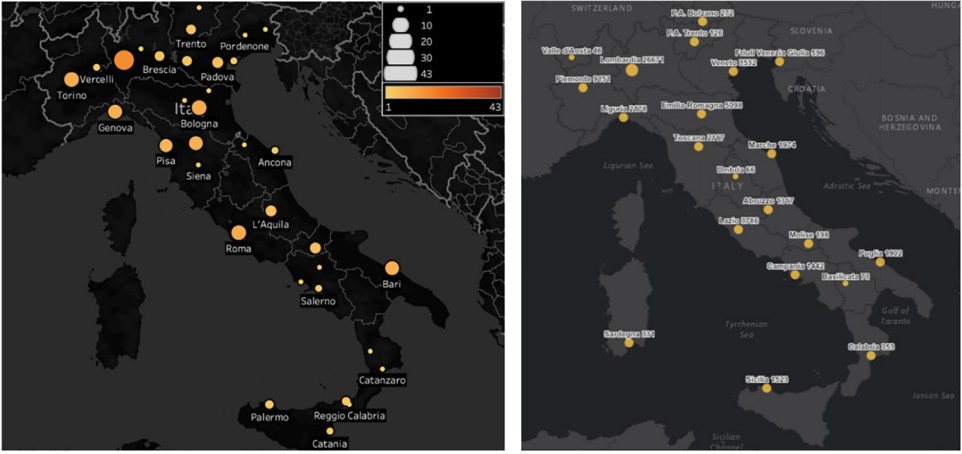

**Fig 5. Comparison between the number of IT initiatives (a) and the number of SARS-Cov-2 cases in Italy on May 2020 [54] (b).**

a given domain with percentages (neglecting the relevance). Scientific domains have been mapped to the ACM classification (see https://dl.acm.org/ccs) as reported in Table 3.

Regarding *high relevant* domains, *Artificial intelligence* is the most frequent (54 occurrences), followed by *Modeling and simulation* and *Medical informatics* (39 activities). This

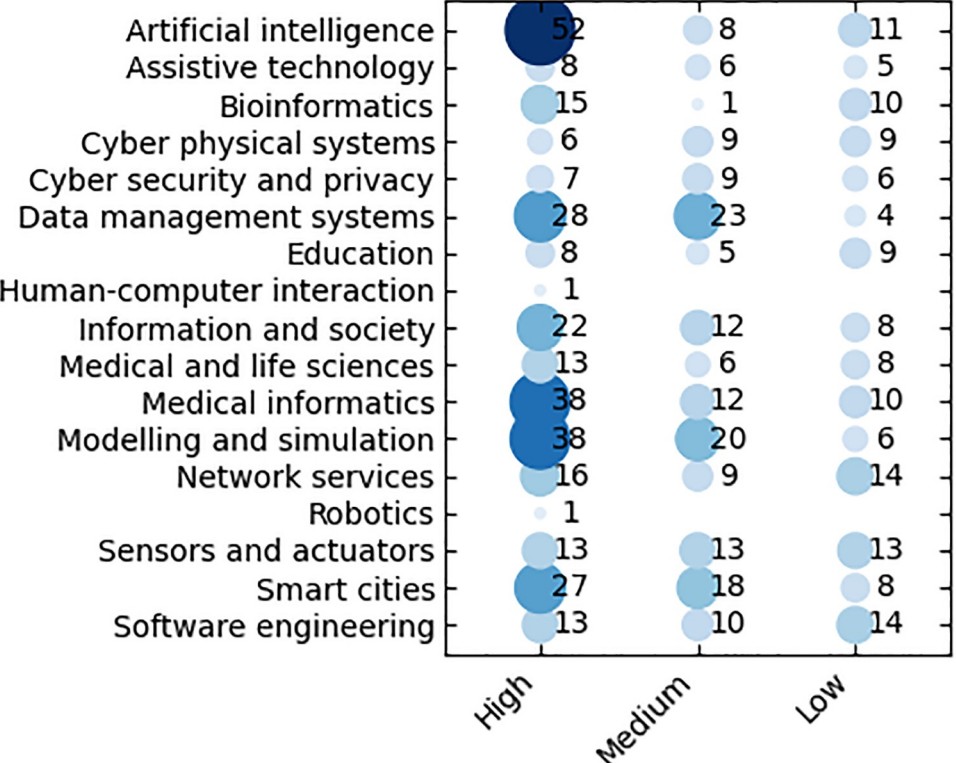

**Fig 6. Number of activities per scientific domains with relevance.**

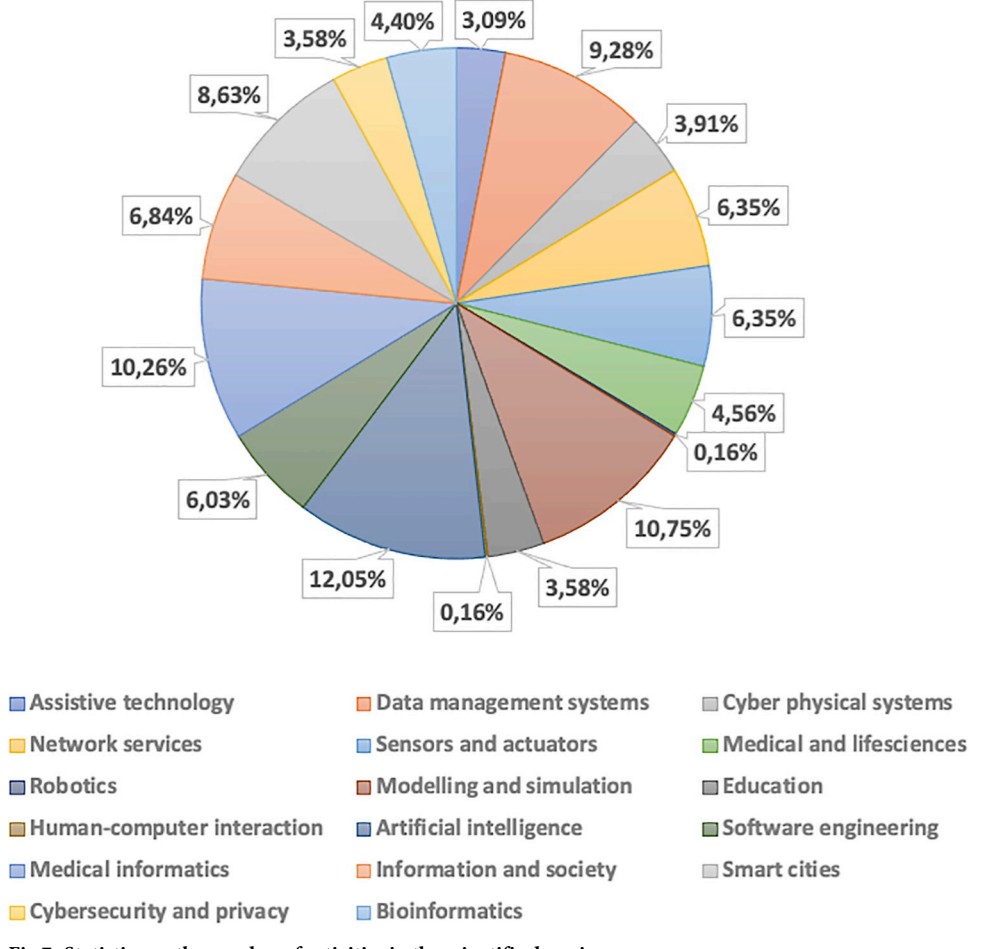

**Fig 7. Statistics on the number of activities in the scientific domains.**

highlights the high suitability of artificial intelligence and modeling and simulation approaches and techniques to deal with Covid-19 related issues (both medical and non-medical), often also overlapped with medical informatics in, e.g., simulating the underling biological phenomena and managing the medical information. The most frequent *medium relevant* domains are those acting as underlying or enabling technologies (*Data management systems* and *Smart cites*) and methodologies *Modelling and simulation*.

Co-occurrences of *high relevant* scientific domains for the surveyed activities, shown in Fig 8, demonstrate high correlation among them, since any scientific domain (excepting *Human-computer interaction* and *Robotics* with only one activity each) co-occurs at least once with any other domain. Considering all the *relevance* levels, indeed, they include at least 5 activities, 3 restricting the scope to *high relevant* activities. Such a descriptive analysis shows that all the research areas are highly correlated. In particular, a strong correlation between *Artificial intelligence* and *Modelling and simulation*, *Artificial intelligence* and *Medical informatics*, and *Data management systems* and *Medical informatics* is highlighted. Thereby, the pattern of adopting computer-based solutions (in particular artificial intelligence and data management) in biological and medical areas emerges clearly emerges from the survey.

**Table 3. Mapping between the systems used in this review to describe scientific domains and the ACM terminology.**

| Scientific domain (ENG) | ACM term |
|---|---|
| Assistive technology | Social and professional/Professional topics/Computing profession/Assistive technologies |
| Data management systems | Information Systems/Data Management Systems |
| Cyber physical systems | Computer Systems Organization/Embedded and cyber-physical system |
| Network services | Networks/Network Services |
| Sensors and actuators | Hardware/Communication hardware, interfaces and storage/Sensors and actuators |
| Medical and life sciences | Applied Computing/Life and medical sciences |
| Robotics | Computer Systems Organization/Embedded and cyber-physical systems/Robotics |
| Modelling and simulation | Computing Methodologies/Modeling and simulation |
| Education | Applied Computing/Education |
| Human-computer interaction | Human-centered Computing/Human computer interaction |
| Artificial intelligence | Computing Methodologies/Artificial intelligence |
| Software engineering | Software and its engineering |
| Medical informatics | Applied Computing/Life and medical sciences/Health informatics |
| Information and society | Applied Computing/Law, social and behavioral sciences |
| Smart cities | Human-centered computing |
| Cyber security and privacy | Security and Privacy |
| Bioinformatics | Applied Computing/Life and Medical Sciences/Bioinformatics |

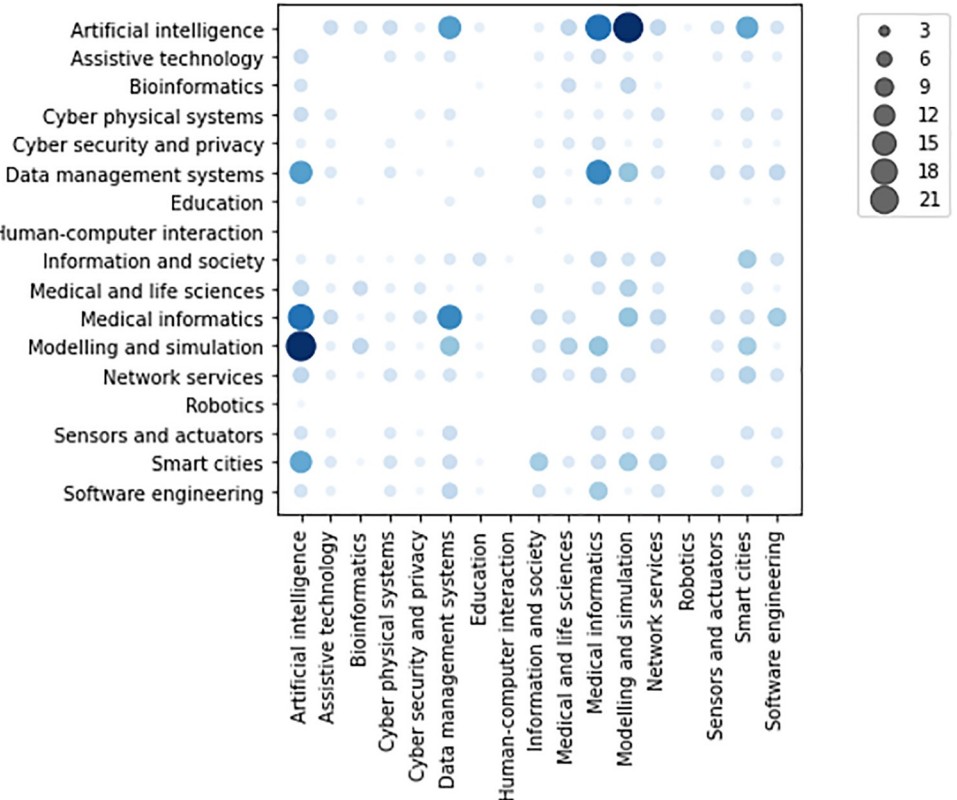

**Fig 8. Co-occurrence of scientific domains indicated as *high relevant* for the surveyed activities.**

## Application context statistics

The surveyed activities have been also categorized according to the application context, considering the 17 categories identified in the previous section qualified into four different levels of *relevance* as above, allowing multiple selections as shown in Fig 9. Related statistics are depicted in Fig 10.

Surveyed initiatives are mainly applied in *Prognostics and diagnostics* (15.5%), *Smart services* (12.5%), *Virology and epidemiology* (12%), *Social services* (10%), *Telemedicine* (8%) and *Social distancing* (7.5%) contexts (as *high relevant*). The 23% of initiatives are associated with a single application context, while 32% of them report two contexts, 22% three contexts and 23% four contexts. The co-occurrence analysis shown in Fig 11 points out an high correlation of surveyed activity application contexts, mainly between *Medical imaging-Prognostics and diagnostics*, *Virology and epidemiology-Prognostics and diagnostics*, and *Smart services-Social distancing*, arising from their similarities.

## Correlation and dependency of categories

The results reported in the previous sections concern single-category analyses. Here, we present an evaluation of the correlation between categories, by means of the co-occurrence descriptive statistics. For each activity, we only considered categories that are classified as *high relevant* for the specific activity.

Co-occurrences between application contexts and scientific domains (Fig 12) show that *Artificial intelligence* is highly related to the Covid19 diagnostics, and thus to *Prognostics and diagnostics* and *Medical imaging*. However, AI is also adopted in *Smart systems* and *Social services*, while *Prognostics and diagnostics* is also related to the *Medical informatics* and *Telemedicine* domains. In biomedical contexts, *Bioinformatics* and *Modeling and simulation* are involved in *Prognostics and diagnostics* and in *Virology and epidemiology*. Almost all scientific

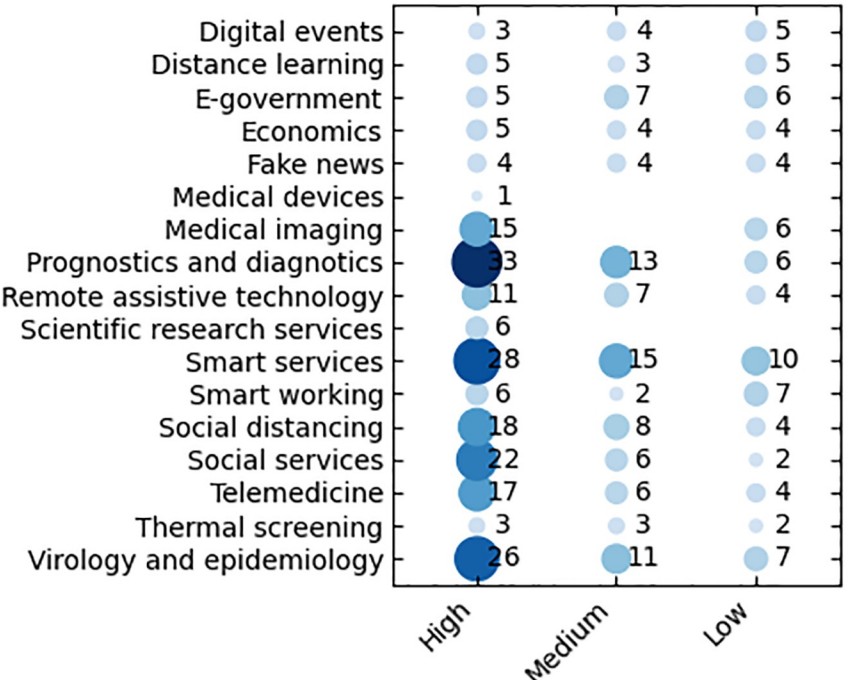

**Fig 9. Number of activities per application contexts with relevance.**

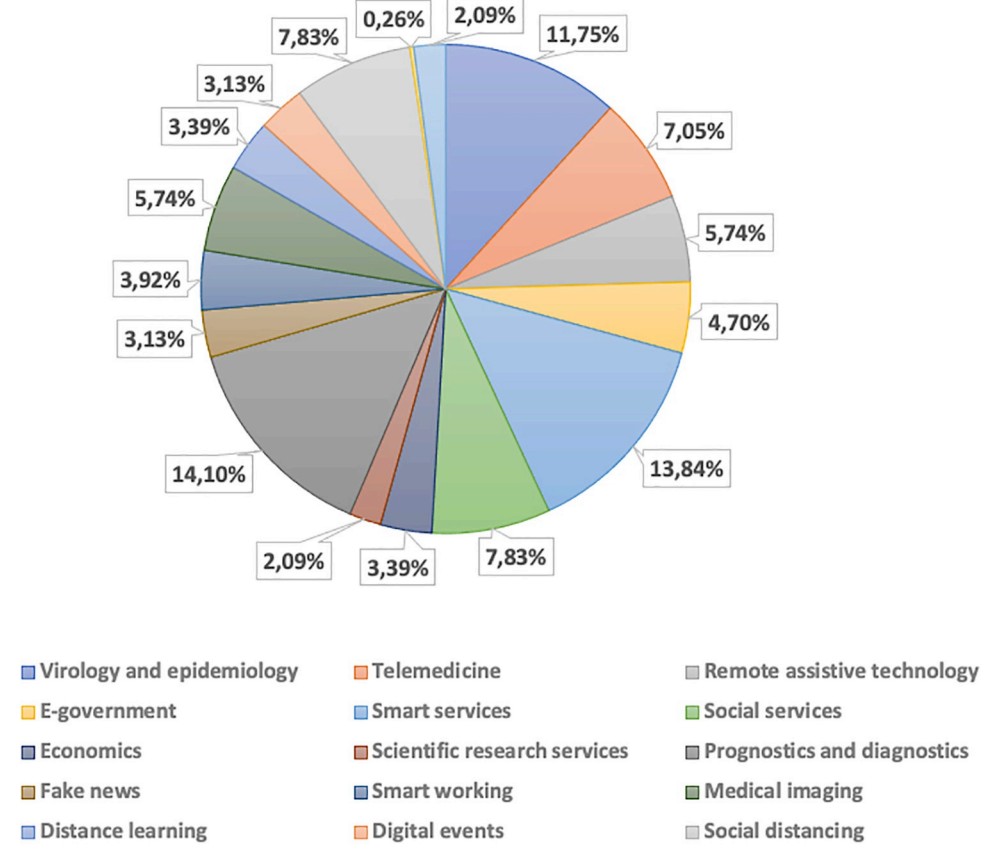

**Fig 10. Statistics on the number of activities per application contexts.**

domains are covered by the initiatives applied at the different geographic scopes. However, *Bioinformatics* activities mainly find application in the international context as *Education* ones, the latter also having applications at a national scope, while all other scientific domains are covered by initiatives at different geographic scopes. Regarding TRL, *Artificial intelligence* is the only domain with activities at all the readiness levels. *Modeling and simulation* and *Smart cites* initiatives span to 8 different TRL. In particular, *Modeling and simulation* initiatives have mainly low TRL, while *Smart cites* ones are in prevalence characterized by higher TRL than the former.

Co-occurrences between application Contexts and geographic scopes (see Fig 13) show that *Prognostics and diagnostics* initiatives have mainly an international scope, as well as *Virology and epidemiology* ones. On the contrary, *Smart services* and *Social services* initiatives span to any geographic scope, with a prevalence in the national scope. *Smart services* and *Prognostics and diagnostics* initiatives span to any TRL. *Medical imaging* and *Virology and epidemiology* ones have low TRL, mainly focused to basic research and prototypes, while *Telemedicine*, *Social services* and *Social distancing* activities have high TRL, closed to 9. The majority of activities having TRL $\geq 5$, indeed, are categorized as *Smart services*, *Prognostics and diagnostics* or *Virology and epidemiology*.

The correlation between the type of activity and the geographic scope is shown in Fig 14. Most of the initiatives implement *Tool*s, applied at any geographic scope. The correlation

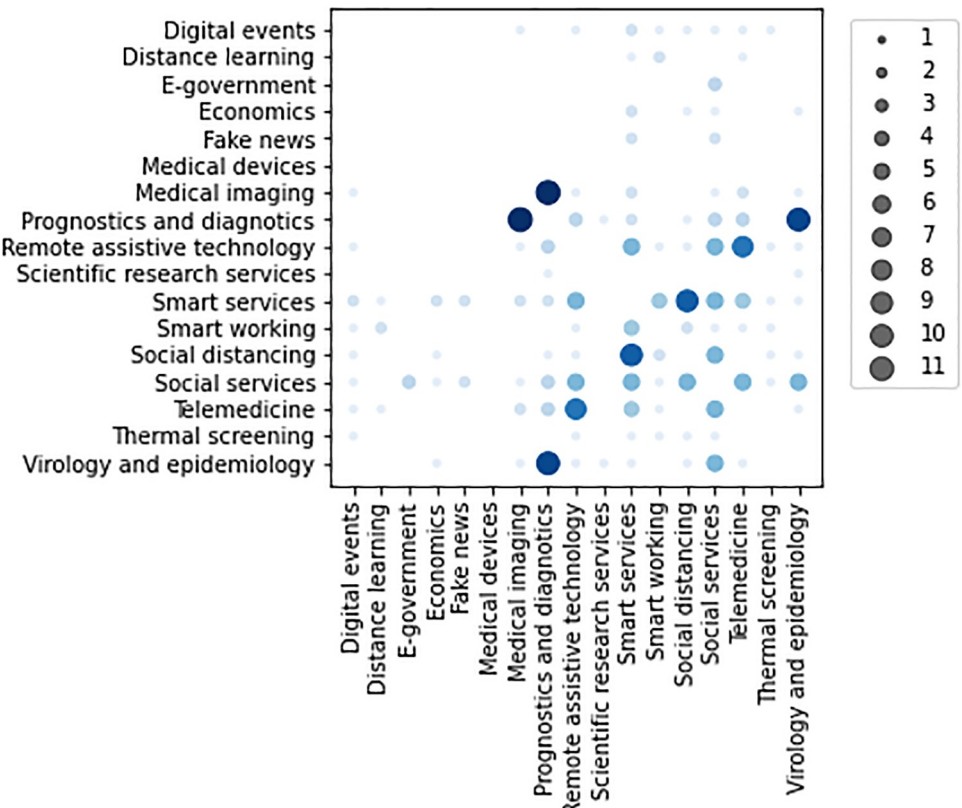

**Fig 11. Co-occurrence of application contexts indicated as *high relevant* for the surveyed activities.**

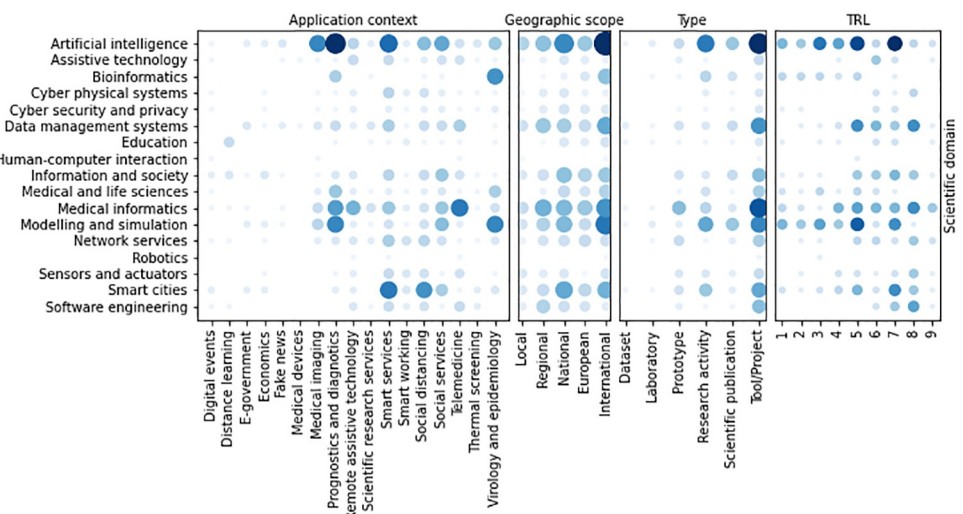

**Fig 12. Co-occurrence of scientific domains with the other categorisations.**

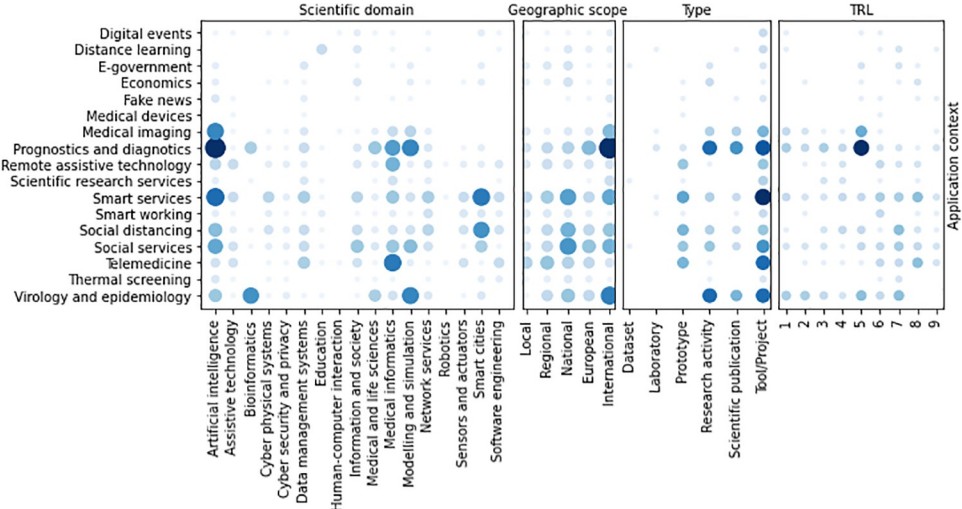

**Fig 13. Co-occurrence of application contexts with the other categorisations.**

between the geographic scope and the TRL is quite evident from the co-occurrence graph: bigger scope initiatives have usually higher TRL than more restricted ones.

The co-occurrence of the type of activities with the other categorizations is shown in Fig 15. The most frequent types tends to higher TRL level. In fact, prototypes and tools have an

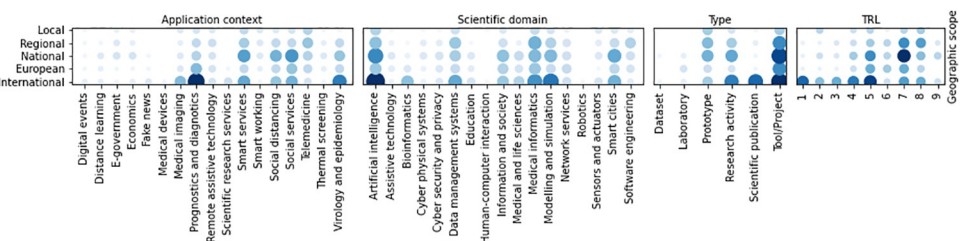

**Fig 14. Co-occurrence of geographic scope with the other categorisations.**

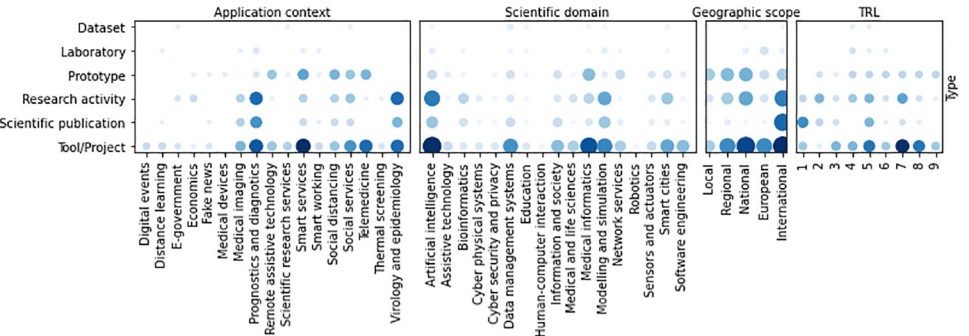

**Fig 15. Co-occurrence of type of initiatives with the other categorisations.**

average TRL of 7. On the contrary, lower TRL levels include, as expected, scientific publications and thematic workshops. research and scientific consulting initiatives cover almost uniformly all TRL levels, excepting level 9.

## Data preprocessing and clustering

After the descriptive analysis, the surveyed initiative data have been further processed (following filtering and cleansing operations) to investigate about hidden correlations not caught above. More specifically, the 128 initiatives have been manually validated, for example to exclude those not containing a proper (e.g too short) description. Thereby, in total, 107 valid initiatives have been selected, thus removing 21 activities from the dataset, while correcting typos, errors and translating terms and sentences in the survey language, i.e. Italian (mainly from English).

The Covid19/IT dataset thus obtained was ready for further elaboration. Specifically, the textual contents of its data items, i.e. the survey initiative descriptions, have been first elaborated by Natural Language Processing (NLP) techniques and then processed by machine learning models to group together semantically similar initiatives. Unsupervised machine learning strategies, in particular clustering methods, have been exploited to such a purpose. They adopt complex data representations as a basis for semantic aggregation operations, demonstrating excellent performance in real-world applications thanks to their generalization capabilities [55, 56]. Approaches based on these principles are quite effective in many data science applications such as marketing analysis, basket analysis, sentiment analysis, prediction, visualization, study of correlations, and tasks of inference of taxonomies and hierarchies. They are therefore a good option for such a research goal, i.e., clustering initiatives by using their textual descriptions to identify correlations among them and then define a reusable taxonomy. Indeed, the main goal of clustering is to group/separate input data into sets with similar features. The groups or clusters thus identified usually show a high similarity score among their data points and a proper degree of separation from those belonging to other clusters.

Among different clustering algorithms, the widely known *K-means* [9] one has been adopted here since it is simple and efficient. To create semantically relevant clustering, a formal representation of textual items, easy to be computed and semantically representative, Is required. The vectorization approach, based on word/token frequencies, can generalize the text representing a document as a set of numerical vectors, one for each of its terms. However, a common term (e.g. an article, a conjunction) is not always a proper indicator of the content of the document, and the word ordering cannot be neglected in semantic representations, pushing towards more complex numerical representations of text such as *"tf-idf"*, Latent Semantic Indexing, Random Indexing, and Page-Rank, to name but a few [57]. Thereby, advanced NLP techniques, including machine translation, text classification, and question answering, take advantage by innovative text representation strategies in terms of performance and reliability.

In particular, a significant contribution was given by distributional semantics models such as word embedding. In this context, Mikolov et al. in [7] proposed the idea that semantically related terms should have similar vector representations. They implemented such an idea by algebraic properties in their vector representation, e.g. the sum of two terms results in a new semantically consistent vector equivalent to the linguistic sum of them: "King—Man + Woman ∼ Queen". Such approaches, as well as Word2Vec [7], Glove [58], and FastText [59], are affected by the problem that multiple concepts associated with the same term cannot be represented, since they correspond to the same word embedding vector (the representation is *context-free*). Moreover, it has been demonstrated that they do not perform well when applied to domains that differs from the one on which they have been trained [60].

New strategies such as ELMo [61], GPT/GPT-2 [62], and BERT [63] overcome this limit by learning a language model for a contextual and task-independent representation of terms. In particular, these models are trained to predict the totality or part of the sentence, e.g. the most probable words preceding and following a specific word in a given domain. Recently, several papers demonstrated the effectiveness of these word embedding techniques in English text NLP, and recently, multilingual models have been distributed.

In this work, different strategies for content representation are compared. In a first attempt, a *word2vec skip-gram model* has been trained by using the surveyed initiative descriptions. Then, a *pre-trained Wikipedia word2vec skip-gram model* [64] and an *Italian version of BERT architecture* (AlBERTo) model [8] have been exploited. The representations thus obtained have been used as input for the K-means clustering algorithm, and the classification results have been then evaluated by a qualitative approach based on t-SNE, PCA, and semantic tags analysis, to visually explain the distributions of the elements among the clusters. The approaches adopted in this paper meet the latest textual content representation strategies proposed in the literature. To the best of our knowledge, their use in Italian textual elements and in the of Covid-19 domain are quite challenging and innovative elements in this context. These techniques turn out to be effective and robust enough to demonstrate the effectiveness of the approach here proposed.

## Clustering models

The clustering process of the surveyed activities is shown in Fig 16 and consists of 4 sequential steps: *NLP pre-processing pipeline*, *vectorization*, *clustering*, and *validation*. The textual descriptions of the initiatives are thus first provided to the *NLP pre-processing pipeline*, which cleans up the text from irrelevant textual elements such as stop words, extremely frequent terms in the collection, and non-alphanumeric characters. It starts from splitting the text into word tokens by the blank space character separator. Each token is then parsed and removed if included in the stop-word list specified by both the NLTK (https://www.nltk.org/) (Natural Language Toolkit) and SpaCy (https://spacy.io/) libraries. In addition, domain-specific stop-words such as "COVID-19", "COVID", "virus", "pandemic" have been also removed as too generic and not much useful for semantic clustering. A further text cleaning step has been applied by removing every token that cannot be traced back to a common Italian word. All non-UTF-8 and UTF-8 characters other than letters and numbers have been removed, keeping only verbs, nouns or adjectives detected by the SpaCy pos tagging algorithm. At the end of the process, the text obtained is similar to the one shown in Fig 17.

The next step in the text cleaning process is the feature selection, focusing on semantically relevant terms identified by the *"tf-idf"* technique. This statistical weighting strategy is based on the idea that a *relevant* term has a high frequency within a specific document and low

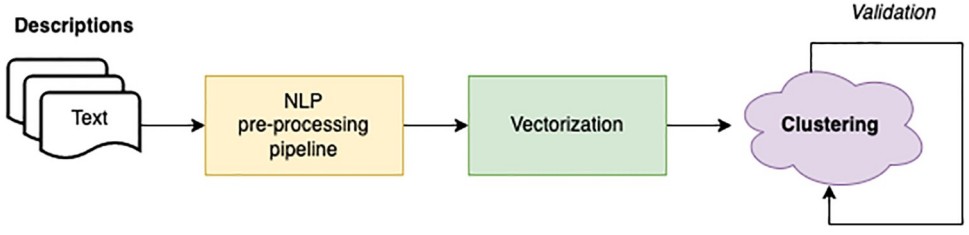

**Fig 16. Clustering pipeline.**

```
1   inter hominem definisce modello validato epidemiologi secondo normative
2   vigenti consente associare tempo reale ciascuna area monitorata indice
3   rischio spazio temporale livello sicurezza dinamica mappa dinamica distanze
```

**Fig 17. Example of pre-processed text.**

frequency in the rest of the collection. The *"tf-idf"* formula is composed by the product of two factors, *"tf"* and *"idf"*, as shown in Eq (2).

$$tf - idf(t, d, D) = tf(t, d) \cdot idf(t, D) \tag{2}$$

$$tf(t, d) = log(1 + freq(t, d)) \tag{3}$$

$$idf(t, D) = log\left(\frac{N}{count(d \in D : t \in d)}\right) \tag{4}$$

The former is *"tf"* (see Eq (3)), i.e. the term frequency, counting the occurrences of the term $t$ in the specific document $d$. The inverse document frequency *"idf"* counts the number of document $d$ in the collection $D$ that contains the term $t$ as shown in Eq (4). It measures the importance of the term in the collection. Each token of the clean text has been ranked by the *"tf-idf"* score, representing each document (i.e. the initiative description) by only the first 100 ranked tokens.

The list of tokens thus obtained is passed to the *vectorization* module, to transform them into numerical vectors. As discussed above, three different models have been adopted to such a purpose: *(i)* a 50d word2vec skip-gram model created from scratch, *(ii)* a 300d word2vec skip-gram model pre-trained on Wikipedia, and *(iii)* a transformer-based BERT model [8]. The first two techniques provide a numerical vector for each token of size equal to that of the embedding model (the source code of the embedding based approaches is available at https://shorturl.at/btDMR). Since the main goal of the vectorization is to cluster the surveyed initiatives, all the vectors corresponding to the same initiative description have been merged into a single vector with average values, thus obtaining a tuple for each initiative. In the latter case adopting BERT, the document embedding vector is obtained by averaging the term vectors observable in the last layer of the deep model (768 dimensions—the source code of the BERT based approach is available at https://shorturl.at/uORUW). The survey initiative vectors thus obtained are ready for clustering.

The K-means unsupervised algorithm has been adopted for clustering the survey initiative vectors due to the low number of data samples and its simplicity. Its main purpose is to categorize $n$ sample points $x_1, x_2, \ldots x_n$ into $k$ clusters by identifying the cluster centroids $c_1, c_2, \ldots, c_k$ and the points within a cluster adopting the *cosine distance* between point vectors as similarity metric. The first step of the K-means algorithm is to choose the number of clusters $k$, which is usually set somehow randomly in the beginning. A proper choice of $k$ implies a prior knowledge of the dataset structure, and can be refined by further runs of the K-means algorithm. The *Validation* process adopted is based on the Davies-Bouldin index (DB), the Sum Squared Error (SSE) and the Silhouette [65] metrics to assess the clustering quality at each iteration, varying the number of clusters from 10 to 20. The DB and the Silhouette methods are related to the concept of dispersion within clusters. For DB the lower value the better, while for Silhouette the higher value the better. The SSE method is a statistical metric that measures the error rate in the clustering by evaluating, for each data point, the distance to the cluster

centroid, thus, as for DB, the lower the better. compact and data points more close to the cluster centroid. Details about the results obtained by validation and clustering steps are reported in the following.

## Experiments and results

The clustering pipeline of Fig 17 has been thus applied to the descriptions of Covid19 initiatives collected by the survey, considering the three different configurations above identified by varying the description semantic textual representation in the implementation as follows:

1. **CONF-1**: adopting word2vec skip-gram for training an embedding distributional space on the initiative descriptions;

2. **CONF-2**: exploiting the word embedding model proposed by Tripodi [64] pre-trained on the Italian version of Wikipedia to ensure high coverage of terms and words included in the model;

3. **CONF-3**: based on the use of contextual embedding, i.e., obtained by the pre-trained AlBERTo model [8].

To produce the *CONF-1* word2vec embedding distributional space and encode the survey descriptions, the Gensim library (https://radimrehurek.com/gensim/) in a Python 3.6 Google Colab environment (https://colab.research.google.com/) has been exploited, setting the replication factor to 5 and the embedding size to 50. Thereby, as described above, a word embedding semantic space where each token is associated with a numerical vector is specified, then ranked by (*tf-idf*), providing the average vector among the most relevant 100 tokens of a description to represent the whole initiative. The resulting 107 vectors (1 per surveyed initiative) are then processed by the K-means algorithm for finding the best value of *k*. By observing the CONF-1 DB, Silhouette and SSE metrics reported in Fig 18, it is possible to identify 14 as a promising value of *k*.

In CONF-2, the embedding distributional space has been trained using a skip-gram strategy, with a 500-element vector, a window of 5, and a replication factor of 10 on the Italian Wikipedia data, obtaining a vector for each surveyed initiative as the average of the top-100 ranked token ones. By observing the CONF-2 DB, Silhouette and SSE metrics reported in Fig 19, the best value of *k* is equal to 12.

In CONF-3, each sentence of the survey initiative descriptions has been given to the AlBERTo transformer model to extract its compact representation (contexual token embeddings) from the last layer of the transformer model deep neural network. As before, the resulting vectors referring to the same initiative description are then merged into an average vector describing the initiative, then processed by the K-means algorithm. The graphs of Fig 20 report the analysis on *k* for CONF-3, showing that *k* = 14 is the value providing low DB, high Silhouette, and low SSE error indicators.

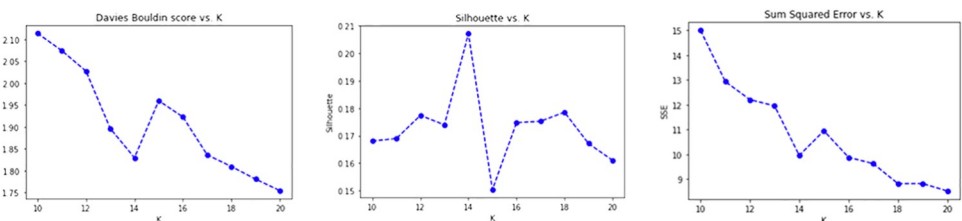

**Fig 18. Validation results obtained for CONF-1.**

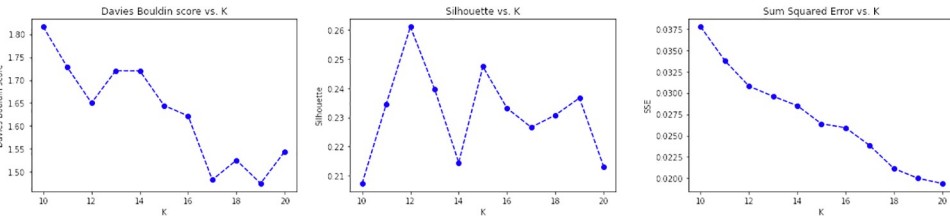

**Fig 19. Validation results obtained for CONF-2.**

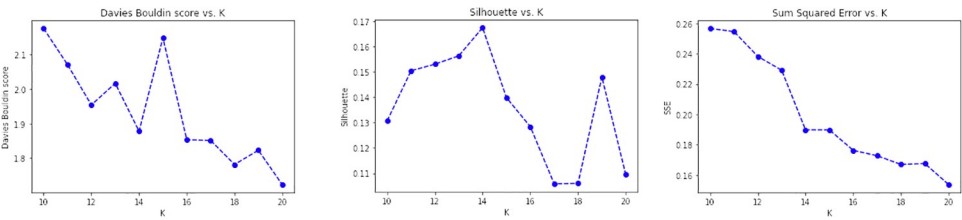

**Fig 20. Validation results obtained for CONF-3.**

Once the best $k$ value for each configuration is identified, the corresponding K-means models have been trained resorting to the scikit learn library (https://scikit-learn.org/stable/modules/generated/sklearn.cluster.KMeans.html), setting $n\_clusters = k$ and $max\_iter = 1000$. Table 4 reports the 107 initiatives clustering results. By investigating the resulting clusters (Figs 21–23), a correlation between the cluster generated and the survey manual classification based on the application contexts can be noticed. In all the configurations, the number of clusters is reduced (from 17 in the manual categorization to 14 for CONF-1 and CONF-3 or 12 for CONF-2), and the resulting clusters mainly group together homogeneously initiatives belonging to the same application context. As an example, CONF-1 cluster 1 includes only prognostics and diagnostics initiatives, as well as cluster 2 (scientific research services), cluster 3 (virology and epidemiology), as well as clusters 4, 6, 8-10, 12, CONF-2 clusters 0, 3-7, 10, and

**Table 4. Distribution of initiatives among the clusters generated by the three configurations.**

| CONF-1 | | CONF-2 | | CONF-3 | |
|---|---|---|---|---|---|
| *Cluster ID* | *# Initiatives* | *Cluster ID* | *# Initiatives* | *Cluster ID* | *# Initiatives* |
| 13 | 16 | 3 | 17 | 5 | 19 |
| 5 | 15 | 1 | 16 | 10 | 15 |
| 1 | 13 | 9 | 13 | 3 | 13 |
| 9 | 10 | 8 | 13 | 6 | 12 |
| 4 | 10 | 11 | 12 | 12 | 10 |
| 0 | 9 | 4 | 9 | 4 | 7 |
| 11 | 7 | 2 | 8 | 2 | 7 |
| 7 | 7 | 6 | 6 | 1 | 7 |
| 2 | 7 | 10 | 5 | 8 | 6 |
| 3 | 4 | 0 | 3 | 9 | 4 |
| 12 | 3 | 5 | 3 | 7 | 3 |
| 10 | 2 | 7 | 2 | 0 | 2 |
| 8 | 2 | | | 13 | 1 |
| 6 | 2 | | | 11 | 1 |

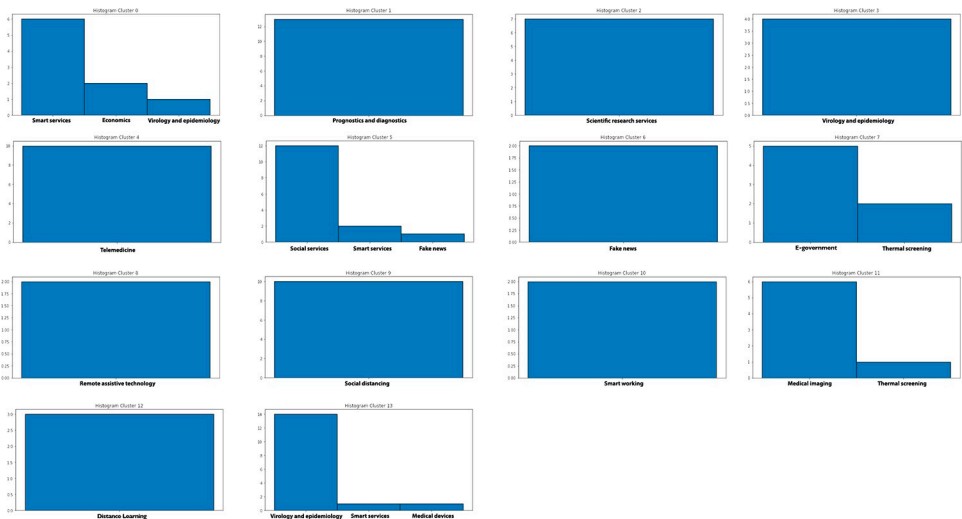

**Fig 21. Details of clusters for CONF-1.**

CONF3 clusters 0, 4-7, 11, 13. Only a few clusters group together initiatives of different contexts, due to the fact of having reduced the number of clusters (from 17 to 14 or 12 in CONF-2), thus redistributing some of the initiatives among the other clusters based on their descriptions.

The main words of each cluster of CONF-1, CONF-2, and CONF-3, are shown in Figs 30–32, respectively, from which it is possible to observe that they are semantically related with the initiative application contexts, as better discussed below. The clusters obtained by the k-means algorithm, mainly those of CONF-1 and CONF-2, are well defined, semantically coherent, and contain only a few initiatives belonging to different application contexts, in particular CONF-2 clusters, *more compact* than CONF-1 ones. Their histograms, shown in Figs 21 and 22, indeed, show how almost all the clusters contain initiatives belonging only to one application contexts, thus creating homogeneous groups, with few exceptions (e.g CONF-1 clusters 0, 5, 7, and CONF-2 clusters 1, 8, and 11).

CONF-3 clustering produces more "noisy" clusters than CONF-1 and CONF-2 ones. This limitation can be traced back to the observable linguistic differences between the lexicon of

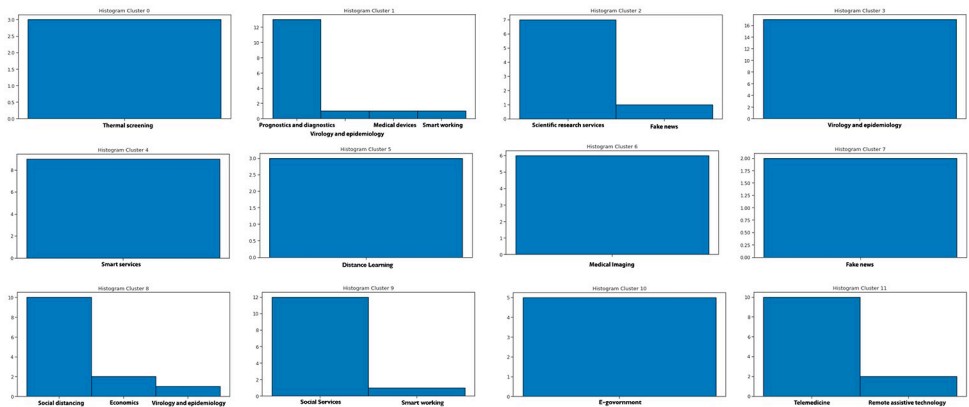

**Fig 22. Details of clusters for CONF-2.**

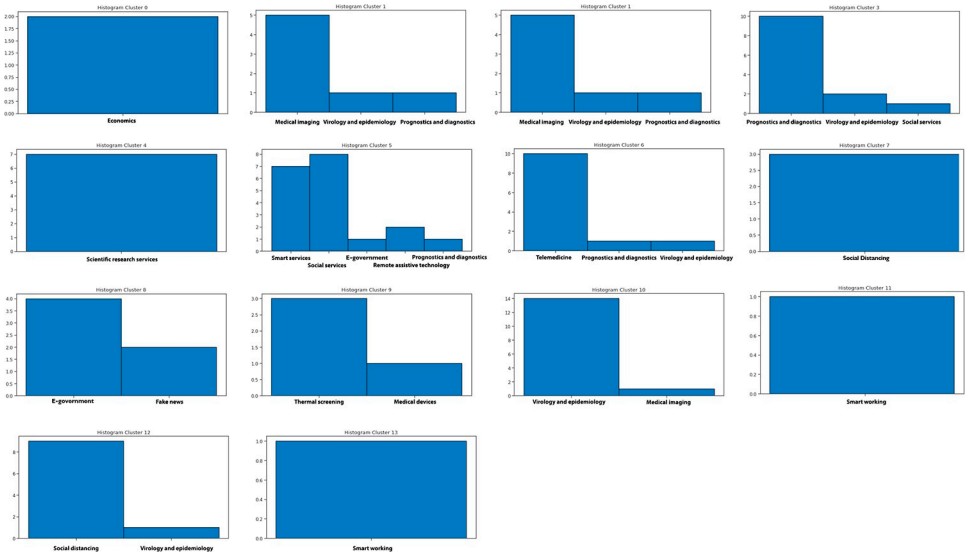

**Fig 23. Details of clusters for CONF-3.**

surveyed initiative descriptions and the one used to train AlBERTo (i.e., tweets), reducing the effectiveness of the vector representations in relation to the application contexts, as confirmed by the histogram plot of Fig 23 and the tags of Fig 32. In cluster 5, for example, initiatives belonging to "Smart services", "Social services", "E-government", "Remote assistive technologies", "Prognostics and diagnostics" are mixed together without a predominant class.

The clusters thus identified allow to obtain a categorization of the surveyed research initiatives into semantically coherent groups. This demonstrates that it is possible to successfully group together initiatives described in Italian on a specific application context by a linear and straightforward pipeline, comparing different textual content representation techniques. In this regard, innovative techniques such as those based on BERT are not enough robust to successfully deal with application domains where complex and never-seen terms arise.

More specifically, to validate the clustering results, state-of-the-art quality metrics [66, 67] have been exploited as benchmarks to compare the (manual) categorization adopted in the survey as the baseline (i.e. the *gold standard*) against the clustering results obtained by the three different configurations above described. Let $P = p_1, p_2, p_3, \ldots, p_z$ be our gold standard clusters $p_i$ ($i = 1, .., z$) and $Q = q_1, q_2, q_3, \ldots, q_{k_o}$ the $o$th clustering algorithm ones $q_j$ ($j = 1, .., k_o$), $N_{i,j}$ is the number of entities in the intersection cluster $p_i \cap q_j$, where $N_i$ and $N_j$ are the cardinality of $p_i$ and $q_j$, respectively. The first score we adopted is the Adjusted Rand Index (ARI) [68], weighting the *consensus* between two sets of clusters. The degree of similarity between these two cluster set can be characterized using a contingency matrix as defined by Eq (5), where $N$ is the number of entities in the dataset.

$$r0 = \sum_{i=1}^{z}\sum_{j=1}^{k_o}\binom{N_{i,j}}{2}, \quad r1 = \sum_{i=1}^{z}\binom{N_i}{2}, \quad r2 = \sum_{j=1}^{k_o}\binom{N_j}{2}, \quad r3 = \frac{2\,r1\,r2}{N(N-1)} \tag{5}$$

$$ARI = \frac{r0 - r3}{0.5(r1 + r2) - r3} \tag{6}$$

The Adjusted Rand Index *ARI* is thus defined as the function of Eq (6), assessing the similarity

**Table 5. Evaluation of the clustering quality.**

|        | ARI     | NMI     | HOM     | COM     | V1      | FMI     |
|--------|---------|---------|---------|---------|---------|---------|
| CONF-1 | **0.85027** | 0.91672 | **0.88815** | 0.94718 | 0.91672 | **0.86522** |
| CONF-2 | 0.84765 | **0.92320** | 0.88595 | **0.96372** | **0.92320** | 0.86458 |
| CONF-3 | 0.45978 | 0.70532 | 0.67325 | 0.74059 | 0.70532 | 0.51001 |

of the two assignments ignoring labels permutations. Perfect labeling is scored 1.0 while imperfect ones are scored lower, by close to zero or negative score. No assumption on the cluster structure is stated by *ARI* thus allowing to compare all kinds of clustering algorithms such as k-means ones characterized by isotropic blob shapes. From the values of Table 5 it is possible to argue that both CONF-1 and CONF-2 obtain *ARI* values close to 1, confirming their quality and closeness to the gold standard annotation, while CONF-3 is less robust. This indicates that the word embedding techniques used in CONF-1 and CONF-2 are good enough to obtain reliable clusters, close to the gold standard.

The Normalized Mutual Information (NMI) [69] is another standard measure for assessing the clustering quality starting from a normalized version of the mutual information measure. The mutual information measure accounts to the "amount of information" that can be extracted from two data source distributions measuring non-linear relations between them. Therefore, a high mutual information value indicates a large reduction of uncertainty whereas a low value indicates a small reduction, while zero means that the two random variables are independent. $Q = q_1, q_2, q_3, \ldots, q_k$ our clustering algorithm results, The *NMI* score is thus expressed by Eq (7)

$$NMI = \frac{I(P, Q)}{\sqrt{H(P)H(Q)}} \tag{7}$$

where $I(P, Q)$ is the mutual information between $P$ and $Q$, while $H(P)$ and $H(Q)$ are the entropy of $P$ and $Q$, respectively. Referring to Table 5, similarly to *ARI* results, CONF-1 and CONF-2 *NMI* values are close to 1, showing high correlations between them and the gold standard annotations, and thus demonstrating the quality of the corresponding clustering results.

Based on the actual annotations of the different designs, i.e., the truth class, it is also possible to perform a conditional entropy analysis. Specifically, referring to [70], both the *homogeneity* (HOM), i.e. the property for a cluster to contain only members of a single class, and the *completeness* (COM), i.e. the property to include in the same cluster all the samples of the same class. The homogeneity evaluates if the cluster data distribution is skewed to a single (gold standard) class. If all the elements of a cluster belong to the same class, the entropy of the cluster data distribution is zero, i.e. the conditional entropy $H(Q|P)$, is 0. Thus, quantifying the homogeneity in the [0, 1] range (1 homogeneous and 0 not homogeneous), it can be specified as *HOM* by Eq (8).

$$HOM = \begin{cases} 1, & \text{if } H(Q|P) = 0 \\ 1 - \dfrac{H(Q|P)}{H(Q)} & \text{otherwise} \end{cases} \tag{8}$$

To satisfy the completeness criteria, the clustering process must assign all elements of a class to the same cluster, defining *COM* symmetrically to *HOM* as specified by Eq (9).

$$COM = \begin{cases} 1, & \text{if } H(P|Q) = 0 \\ 1 - \dfrac{H(P|Q)}{H(P)} & \text{otherwise} \end{cases} \tag{9}$$

The *V1* metric combines homogeneity and completeness into a summary score defined by Eq (10),

$$V1 = \frac{(1 + \beta) \times HOM \times COM}{\beta \times HOM + COM} \tag{10}$$

where, by default the value of $\beta$ is equal to 1.0. As highlighted in Table 5, CONF-1 and CONF-2 are highly homogeneous and complete, in particular CONF-2 showing the highest V1 score (0.92320).

The Fowlkes-Mallows index (FMI) is a performance metric assessing the similarity of clusters obtained through different clustering algorithms, providing absolute values when compared to the ground truth, i.e. the perfect baseline cluster. The Fowlkes-Mallows index *FMI* is defined as the geometric mean of the pairwise precision and recall measures defined by Eq 11,

$$FMI = \frac{TP}{\sqrt{(TP + FP)(TP + FN)}} \tag{11}$$

where *TP* is the number of True Positive (i.e. the number of pair of points that belong to the same clusters in both the true labels and the predicted labels), *FP* is the number of False Positive (i.e. the number of pair of points that belong to the same clusters in the true labels and not in the predicted labels) and *FN* is the number of False Negative (i.e the number of pair of points that belongs to the same clusters in the predicted labels and not in the true labels). The score ranges from 0 to 1, and high values indicate high similarity between the clusters. The FMI values of Table 5 confirm the above results, with CONF-1 and CONF-2 providing the best clustering quality ($\sim 0.865$). This demonstrates the effectiveness of the word2vec-based content representation approach while the BERT one failed to be effective in this specific application scenario. This, however, does not limit its possible use in application domains where a more comprehensive coverage of the vocabulary used in the text is available.

## Cluster visualization and interpretability

A deeper investigation on the initiative distribution among the clusters obtained by the 3 configurations above discussed can be performed by the t-Distributed Stochastic Neighbour Embedding (t-SNE) [71] and Principal Component Analisys (PCA) [72] visualizations. In particular, as shown in Figs 24–26, the t-SNE feature reduction approach allows to obtain a clear visualization of the identified clusters, compact enough to be considered as a quite readable outcome for further analysis. On the other hand, PCA results, shown in Figs 27–29, do not allow to visually identify clusters in none of the 3 configurations of the experiments. From these it is possible to argue that the clustering features cannot be linearly reduced as done in PCA, but have non-linear relationships that are better caught by the t-SNE analysis.

Furthermore, for the sake of interpretability, as discussed above, an analysis on the most frequent terms of each clusters has been performed to highlight their correlations with application contexts, through the tag-clouds shown in Figs 30–32 (unfortunately in Italian). From these, it is possible to argue that a correlation is almost always observable, especially for the

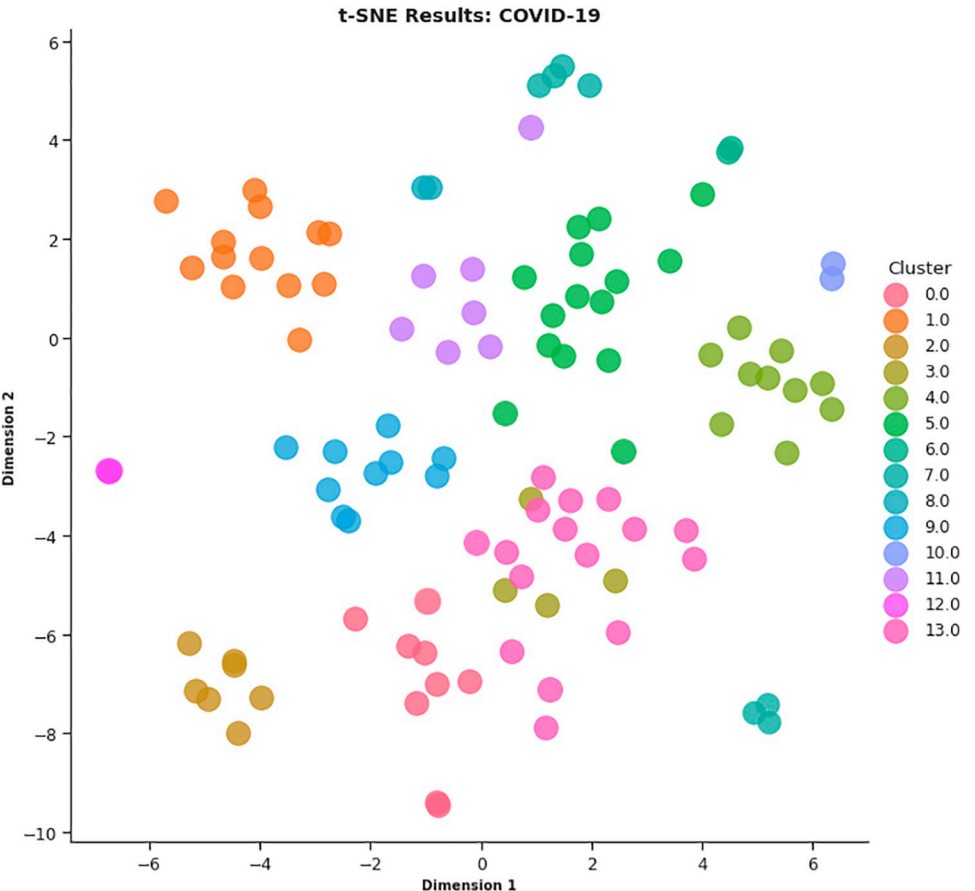

**Fig 24. t-SNE visualization of clusters for CONF-1.**

clusters of CONF-2, visually confirming the above semantic analysis results of Table 5, i.e. CONF-2 is the best candidate for the final taxonomy step. Specifically, in CONF-2 cluster 1, for example, the most frequent application context is "Thermal screening", and among the words most frequently used in the corresponding initiatives there are terms such as "temperature", "behavior", "position", "thermal imaging camera", "software". Similar results can be observed in each of the 14 clusters of CONF-1 and, less clearly, in those of CONF-3. In conclusion, CONF-2, based on word embedding from Wikipedia, proved to be the most effective one, allowing to obtain a more homogeneous and complete cluster classification in 12 well distinct categories with semantically similar initiatives versus the original one based on 17 categories/application contexts.

Considering the CONF-2 results as the reference ones for further investigation, at first it is possible to argue that the generation of clusters allowed to highlight relationships quite challenging and to be identified otherwise. In particular, such correlations can be inferred by analyzing initiatives that fall into a different cluster from the one originally identified by the survey phase. In the CONF-2 cluster 1, there are initiatives mainly coming from the original "Prognostics and diagnostics" category, as well as some originally labelled "Virology and epidemiology", "Medical devices" and "Smart working". However, for example, the "Computational Virology and Epidemiology" initiative that has been categorized as "Prognostics and diagnostics", touches topics closed to them of "data mining for epidemiological prediction",

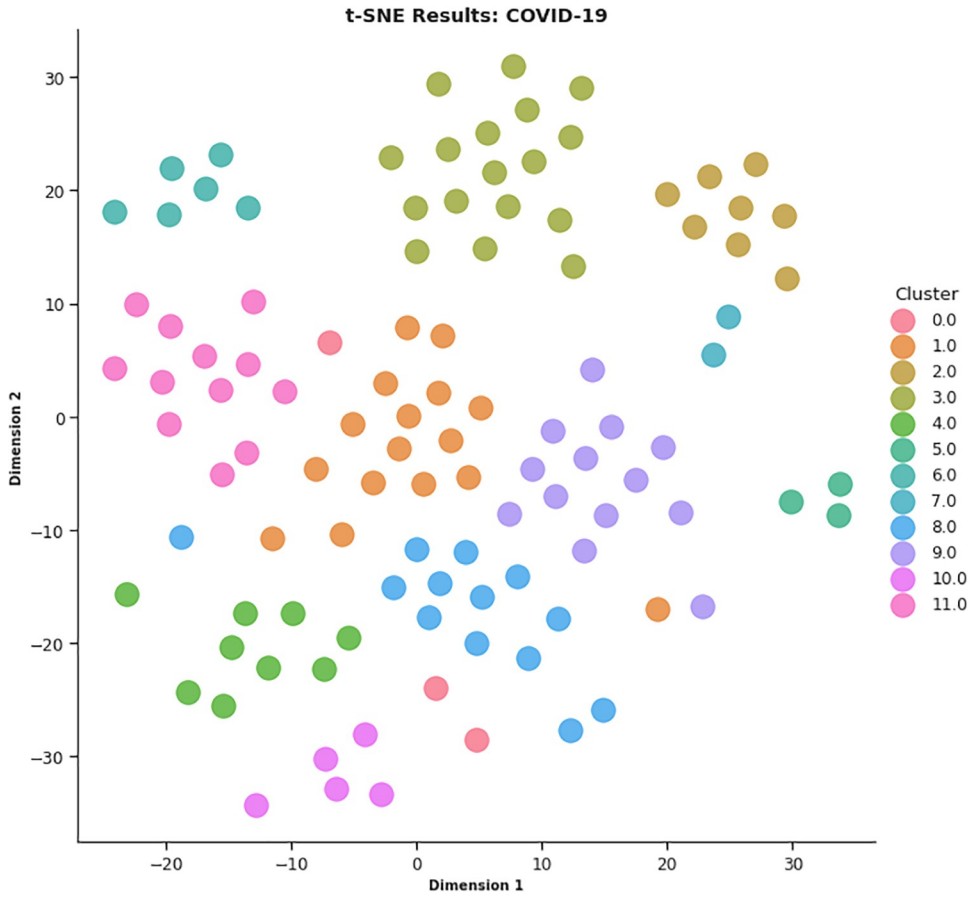

**Fig 25. t-SNE visualization of clusters for CONF-2.**

thus it is possible to consider it as a borderline wrong classification. The "Medical Devices" initiative categorized as "Prognostics and diagnostics" by the CONF-2 clustering is related to the design of an Intelligent Mask, thus also linked to prognosis and diagnosis, in line with the clustering proposed by CONF-2. In the context of "Smart Working", a "smart digital contact tracing" solution has been categorized by CONF-2 as "Prognostics and diagnostics", since its description often presents terms such as "detection", "diagnosis", and "positive".

Similarly, in CONF-2 cluster 2 there is an initiative in the "fake news" area, while most cluster members are related to "services for scientific research". It is well known that scientific research is based on scientific articles and datasets that are the core of the cluster topics. The initiative erroneously included in the fake news cluster concerns a research competition organized on such topic, i.e. fake news. There is therefore a strong correlation between this initiative and the "services to support scientific research" topic. The same pattern is observable in clusters 8, 9, and 11. It is thus possible to argue that the qualitative analysis of CONF-2 clusters has allowed to observe that often the initiatives incorrectly classified in a cluster are related to more than one application context. This cross-dimensionality allows to understand better the issues addressed by the collected initiatives thus identifying some of them as borderline and potential members of different clusters. Therefore, it is possible to conclude that it is not possible to obtain a robust clustering in a rigid scenario, but by adopting different aggregations of

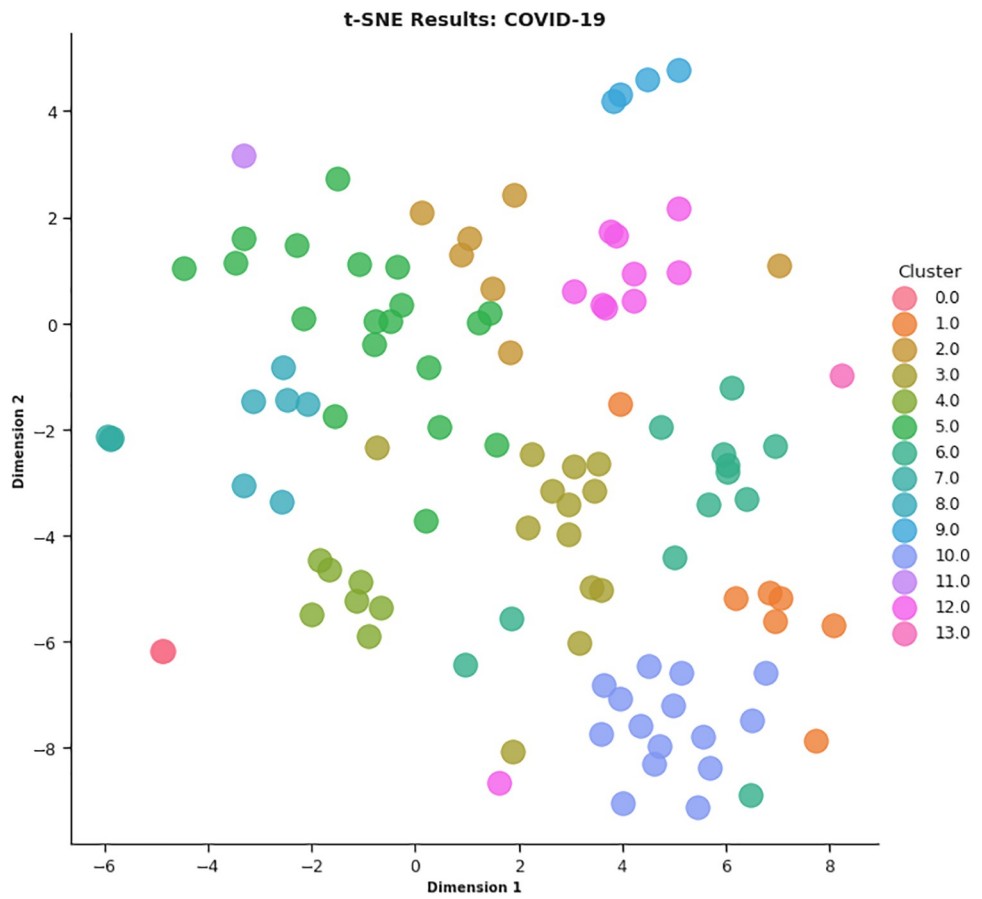

**Fig 26. t-SNE visualization of clusters for CONF-3.**

the 12 CONF-3 clusters, it is possible to represent all the surveyed initiatives in a complete and transversal way.

## Taxonomy

The last step of this journey into the Covid19 IT (ITalian-Information Technology) research is the taxonomy. After collecting, analyzing, preprocessing and clustering the survey response dataset, based on the outcomes from the clustering (CONF-2), a taxonomy of the research efforts on Covid19 from the Italian community on Information (Science and) Technology is specified. As discussed in the related work section, a plenty of models, taxonomies and ontologies have been defined so far in the different areas of Covid19 researches. Among them, the ontology proposed in [2] emerges as one of the most interesting attempts since it i) has been obtained from a real dataset adopting the well-known *case-based reasoning* technique for automatic taxonomy generation, identifying a quite large and detailed ontology (196 classes and 459 axioms), ii) is not exclusively focused on a specific area, but covering several aspects and different perspectives, including the research one, and iii) is publicly available (https://github.com/NathanielOy/covid19ontologies). Thus, following best practices in ontology development, i.e. reusing existing artifacts and models, the Covid19/IT ontology has been obtained by just extending the one proposed in [2] as shown in Fig 33. In particular 13 classes (209 classes

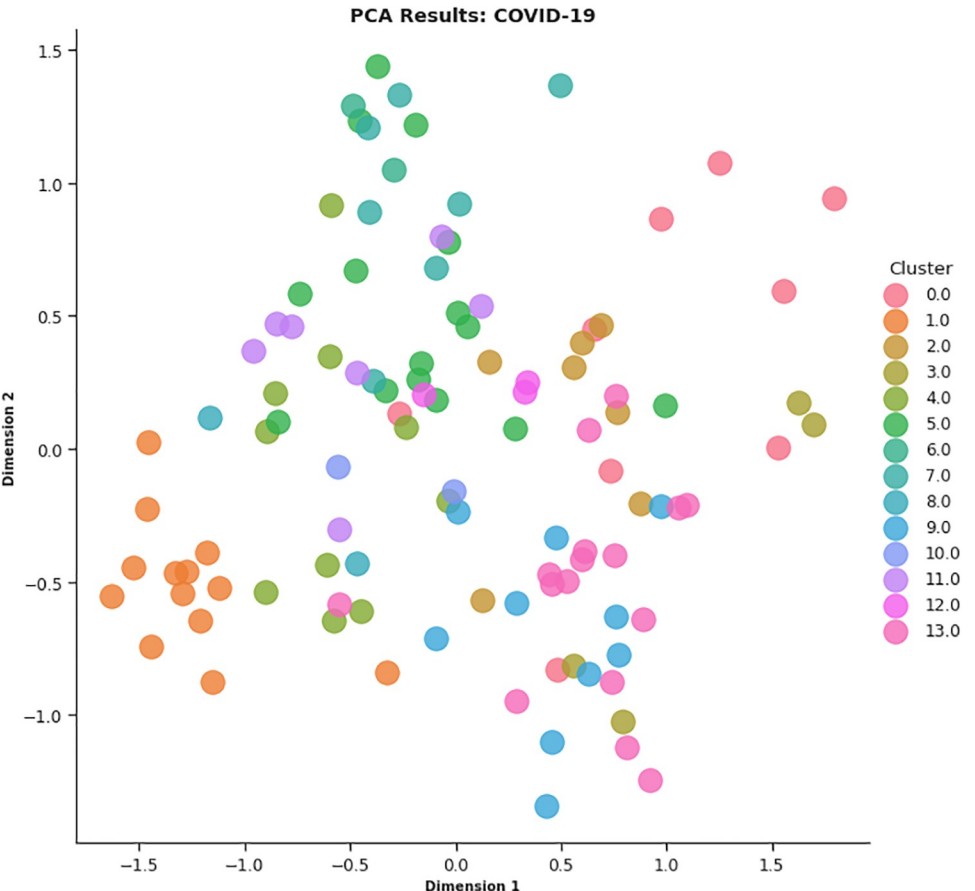

**Fig 27. PCA visualization of clusters for CONF-1.**

in total) and 51 axioms (510 in total) have been added to the original ontology. To such a purpose, the above referred ontology OWL model (cbr_covid19.owl) has been modified by exploiting the *protégé* tool [73].

More specifically, Fig 33 shows a collapsed view of the overall taxonomy expanding only the new classes, which extend ResearchAndLearning as ComputerBasedResearch (highlighted in Fig 33) and its subclasses. As above discussed, these have been obtained by the clustering process described in Section Model, considering CONF-2 as baseline and thus identifying 12 subclasses for ComputerBasedResearch. In some cases such subclasses implement multiple inheritance from other ResearchAndLearning children than ComputerBasedResearch. Referring to the 12 CONF-2 clusters of Fig 22, these have been defined as follows: ComputationalVirologyAndEpidemiology corresponding to Cluster 3, also specializing ResearchOnEpidemiology, ResearchOnMathematicalModelling, ResearchOnVirology; ComputerAidedPrognosticsAndDiagnostics corresponding to CONF-2 Cluster 1 while extending ResearchOnDiagnostics, ResearchOnTreatment, ResearchOnVirology; DistanceLearning corresponding to Cluster 5; EGovernment corresponding to Cluster 10; Infodemic corresponding to Cluster 7 also extending ResearchOnInfectionControl, ResearchOnInfectionPrevention, ResearchOnRiskCommunication; MedicalImaging corresponding to Cluster 6 and specializing ResearchOnDiagnostics, ResearchOnTreatment,

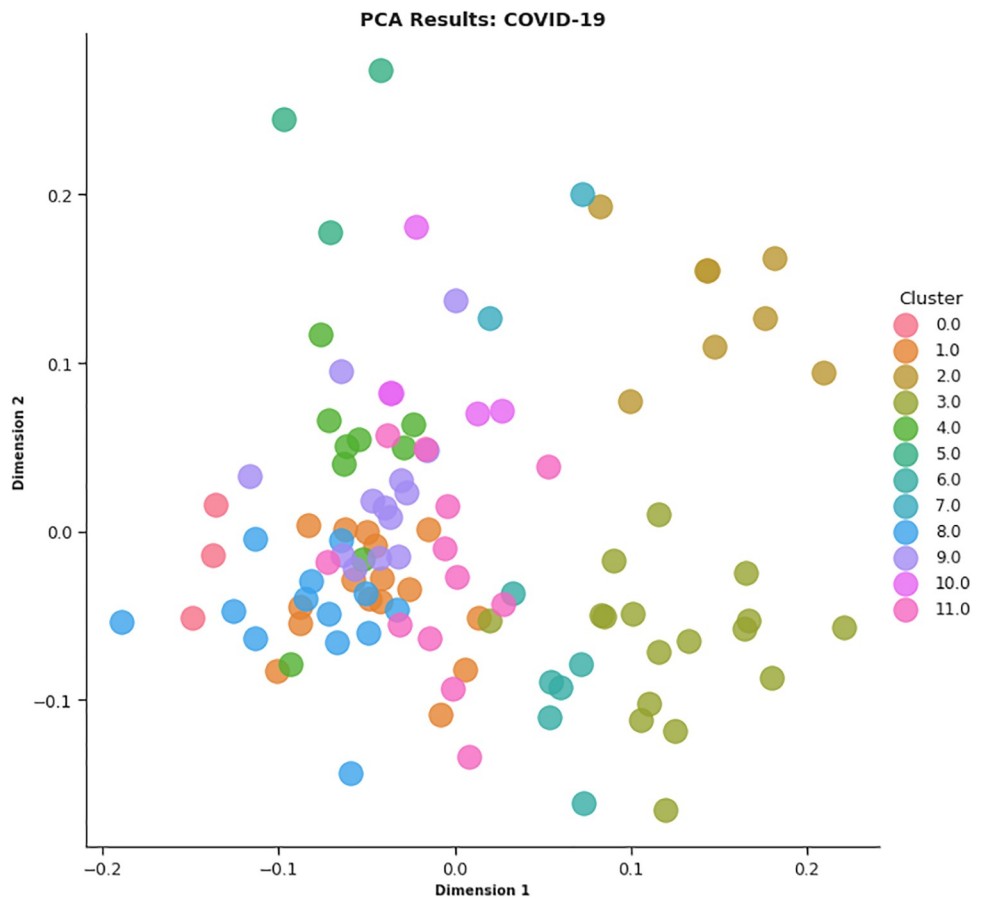

**Fig 28. PCA visualization of clusters for CONF-2.**

`Surveillance`; `ResearchServices` corresponding to Cluster 2; `SmartServices` corresponding to Cluster 4 and also extending `Surveillance`; `SmartWorkingAndSocialServices` corresponding to Cluster 9 while extending `ResearchOnClinicalCare`, `ResearchOnRiskCommunication`; `SocialDistancing` corresponding to Cluster 8 while extending `ResearchOnInfectionControl`, `ResearchOnInfectionPrevention`, `Surveillance`; `TelemedicineAndAssistiveTechnologies` corresponding to Cluster 11 also specializing `ResearchOnInfectionControl`, `ResearchOnInfectionPrevention`, `ResearchOnTreatment`, `Surveillance`; and `ThermalScreening` corresponding to Cluster 0 while extending `ResearchOnDiagnostics`, `ResearchOnInfectionControl`, `ResearchOnInfectionPrevention`, `Surveillance`.

Thereby, the Covid19/IT ontology thus obtained improves one of the most interesting Covid19 ontology [2] with new concepts from the IT area, thus aspiring to be a reference model for future work, also in technical contexts. As discussed above, indeed, existing ontologies are mainly focused on medical concepts and contexts, with exceptions related to specific technologies (mobile health, networking etc.). In this light, the proposed ontology could be a way to group together all such efforts, providing semantic indexes to retrieve Covid19 IT-related information. The Covid19/IT ontology here proposed has been published in an online repository (https://github.com/marcopoli/covid19_clustering).

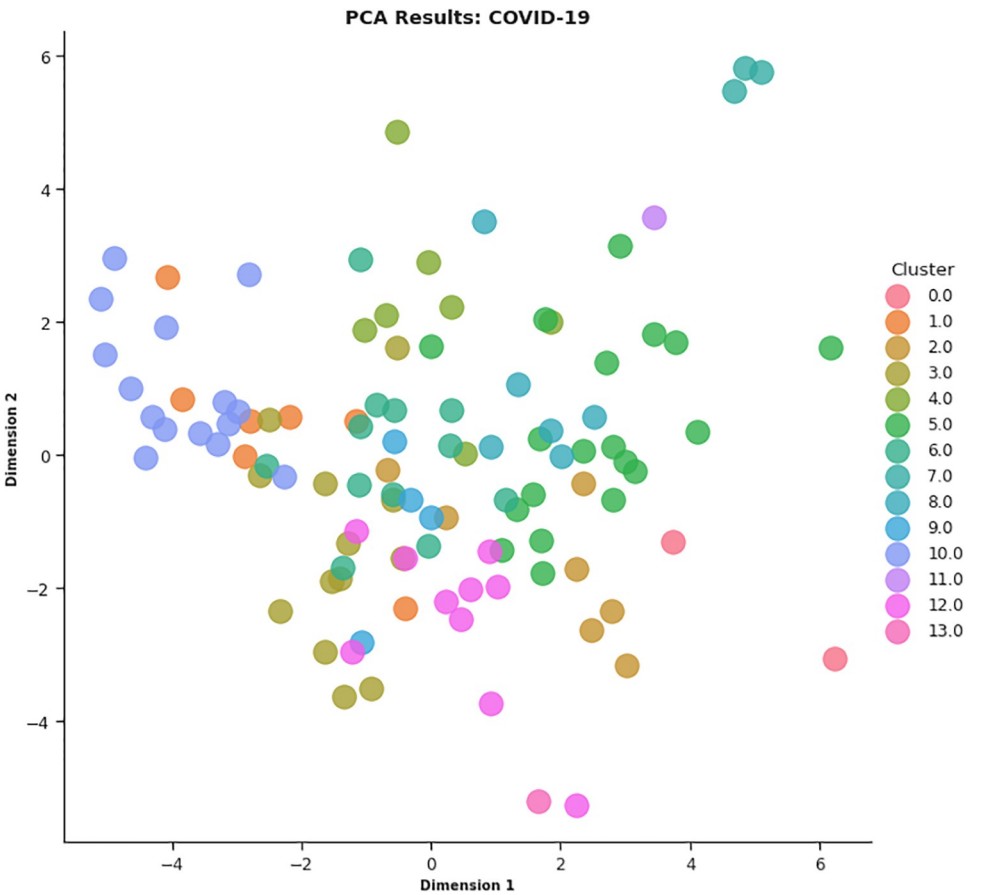

**Fig 29. PCA visualization of clusters for CONF-3.**

## Lesson learned, conclusions and future work

In this paper, the results of a survey about the activities of the Italian Information Science and Technology community against Covid19 developed during the "first wave" in Italy (Jan-May 2020) have been presented, discussed and further elaborated. Interesting findings from both the analysis of the survey initiative dataset and its further processing through NLP-driven clustering techniques emerged into a taxonomy for Covid19 IT researches and solutions. In particular, it can be observed that the response to the first Italian pandemic wave has been quite uniform and prompt, as also emerged from similar surveys. The most significant number of activities is located in large universities and research centers of large cities such as Milan, Rome, and Turin. Great part of activities provide concrete tools and solutions to the real needs and problems due to Covid19, in different contexts, to support the diagnosis, treatment, and prevention of the disease, as well as to deal with Covid19-related social and economical issues through the application of innovative digital technologies and solutions. On the other hand, the scientific community provide tools such as scientific reports, taxonomies and datasets published in competitive national and international journals. Moreover, most of the surveyed solutions can be applied at the international level, demonstrating that they are highly scalable and interoperable.

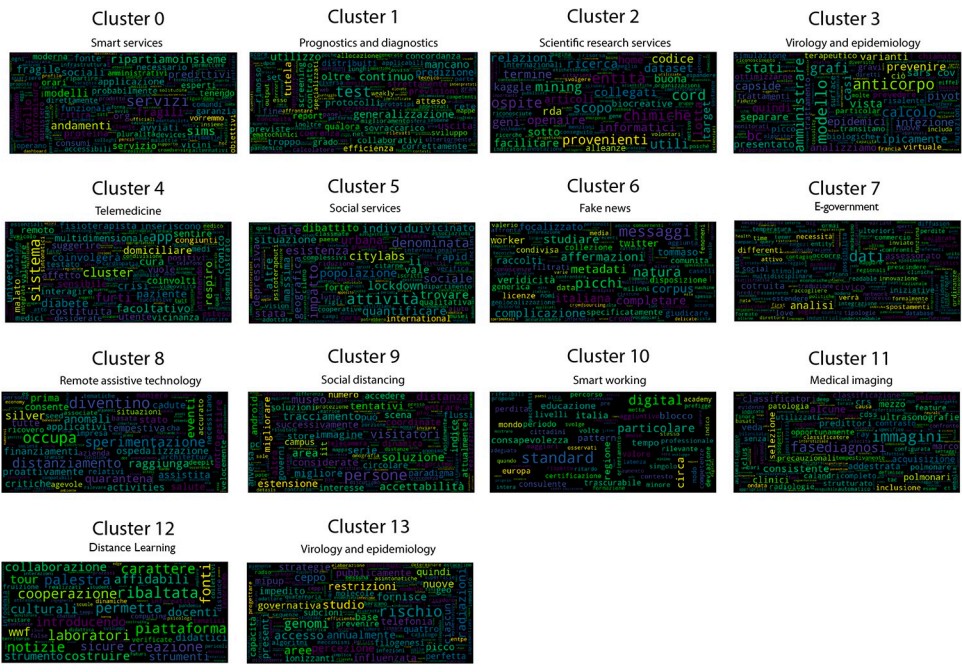

**Fig 30. Tag-cloud of clusters for CONF-1.**

These results mainly highlight the need of tools like the survey here discussed to disseminate and inform the scientific community and all other stakeholders (e.g. politicians, companies, public administrations) about existing solutions, methodologies, datasets. It is necessary to *share* knowledge, skills, experiences, and technologies to enable retrieving and reusing ready-made available solutions avoiding to reinvent the wheel that in emergency may introduce extremely risky delays. This implies to revise existing policy, even to rethink them, switching towards proactive solutions that can prevent epidemics and pandemics. At the same time, more effective reactive solutions for pandemics containment should be implemented.

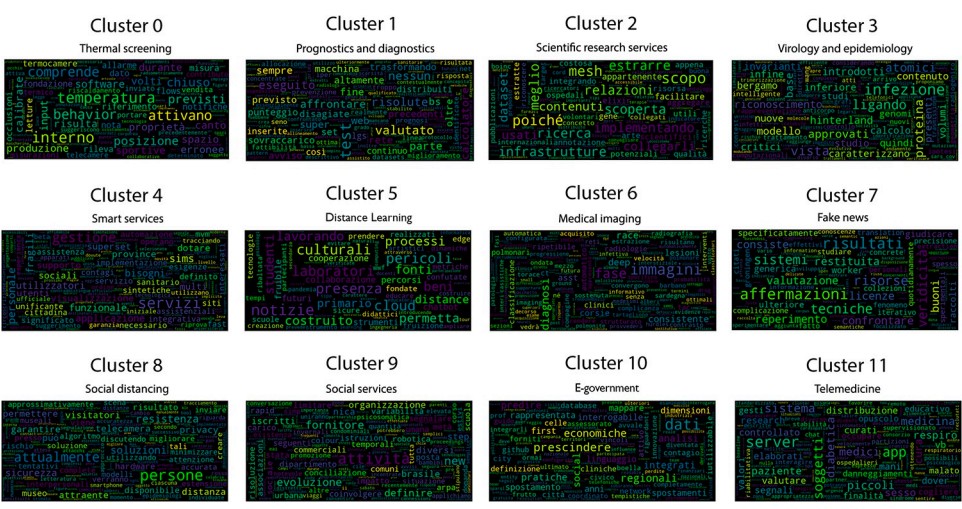

**Fig 31. Tag-cloud of clusters for CONF-2.**

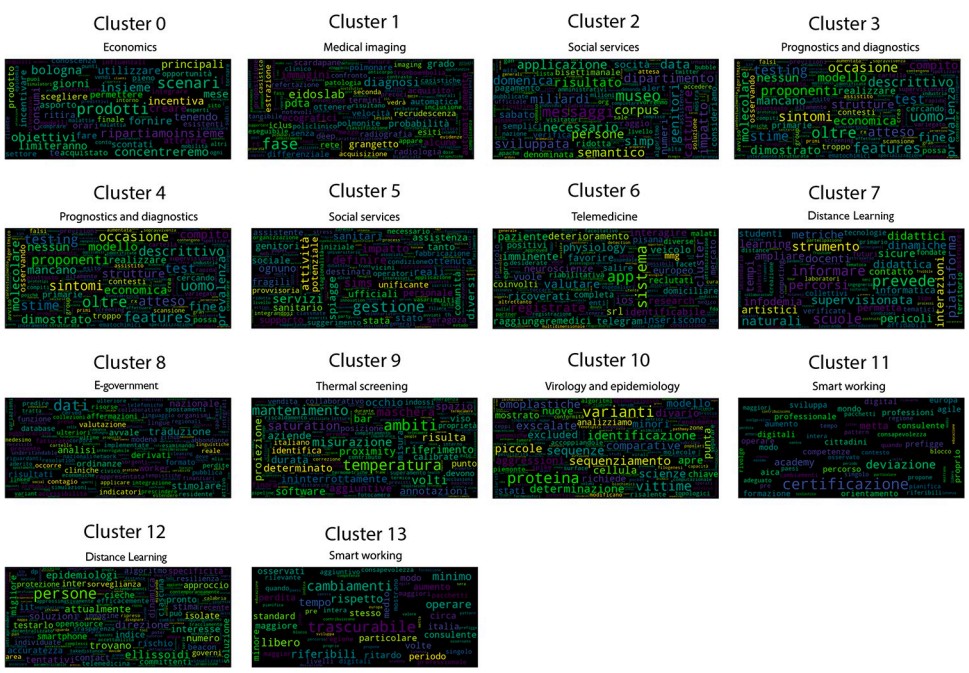

**Fig 32. Tag-cloud of clusters for CONF-3.**

The solutions here surveyed can be just the basic tools, the mechanisms, the bricks by which current administrations and governments have to build pandemic prevention and containment plans and policies towards a resilient world. In this context, the main contribution of this paper is a way, a methodology, a process for implementing such kind of surveys, going beyond

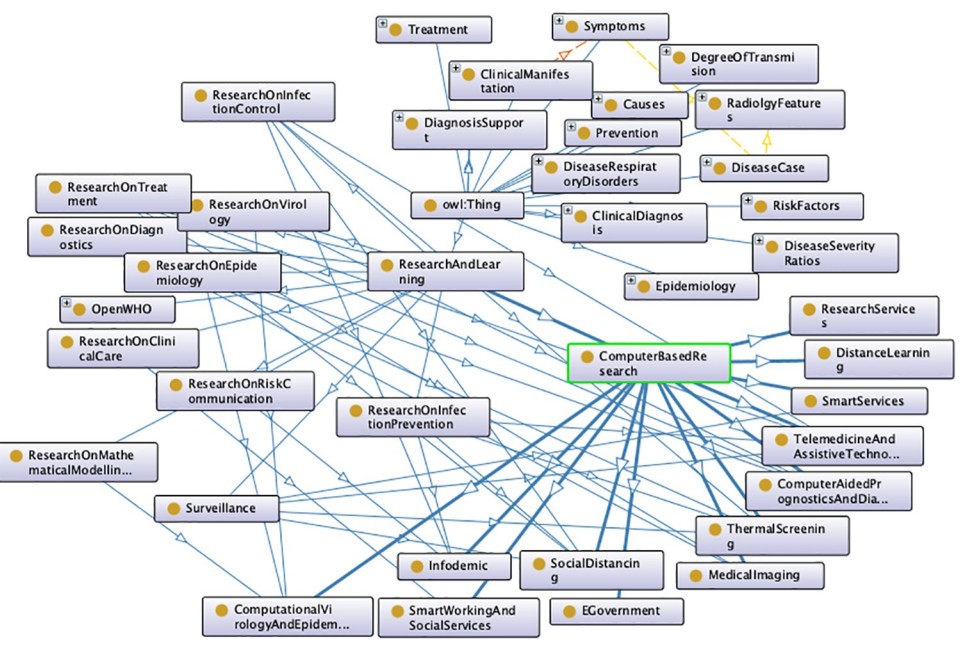

**Fig 33. Taxonomy.**

data collection, cleansing and sharing of a dataset, with in depth analyses that allow to generalize the findings into, e.g., taxonomies or ontologies to support future investigations.

With regard to IT, its strategic importance in any Covid19 application context and domain is quite evident and has been also highlighted by the survey results. However, proper infrastructure and services for sharing (and ingesting, managing, curating) health, medical and clinical information, such as centralized storage systems, DBs, till bio-banks are required. Covid19 calls for more effective applications to deal with pandemic issues, on epidemiology, virology and other medical/clinical contexts, as well as in economical and social ones. Existing solutions could and should be improved for, e.g., achieving better and faster diagnoses and prognoses, or making efficient the digital contact tracing by rethinking specific apps, one of the biggest (epic) failure of the IT community worldwide against Covid19 [74]. The whole system and service engineering needs to be rethought, as suggested in [3], also taking into account resilience and sustainability, not anymore only efficiency, when developing products, systems and services.

From a technical standpoint, a fundamental step in this process is to remove stopwords and too common terms, not enough relevant for or even affecting the clustering quality by introducing noise. The number of clusters is quite important to obtain meaningful results, and therefore the use of available state-of-the-art metrics to measure the intra-clusters coherence is always recommended. Regarding the text representation, the adoption of embedding techniques is somehow mandatory. When few data are available, it is better to rely on pre-trained models, possibly on data similar to the application context ones. The clustering process results allows to claim that such an approach is general enough to be reused in similar application areas. They confirmed the original (manual) classification, restricting from 17 to 12 categories. This allowed to validate the proposed model and thus transfer such knowledge into a public taxonomy on IT Covid19 activities released as one of the contributions of this work for future investigations.

In spite of the valuable insights offered in this study, it also possesses some limitations that can be considered challenges for further research. A significant limitation is represented by the available dataset (only 128 samples collected) therefore the need to have more training data arises to improve the overall accuracy of the clustering algorithm. Moreover, a further limitation is the brief description of the initiatives, which in some cases miss key details. It is therefore important to adopt a formal and well-defined protocol for a proper and correct observation of correlations between activities in the clustering approach. In particular, textual descriptions must be well focused and long enough to provide significant results: some of the surveyed initiative descriptions, indeed, were too brief or not well focused, including details regarding funding or other research activities. To identify a set of key elements and features that properly characterize an initiative, it at least a two/three paragraph description is required. Another limitation is related to the Italian: most of th NLP systems technologies and models are based on and optimized for the English language, better capturing key features while avoiding overfitting problems.

As an extension of the work here proposed, we plan for a new survey to observe how many of the previously surveyed activities have been completed and actually used in real applications. It would also allow to investigate the dynamics of IT-related Covid19 activities, if there would be any seasonality or specific "winner" trend, technology, discipline and application domain. A way to implement the Covid19 activity monitoring could be by establishing a *permanent observatory* on such a topic, hopefully multi-/inter-disciplinary, but with a strong focus on digital aspects of Covid19, a quite challenging mid-term goal for our future work. In this direction, an interesting idea could be to establish a multidisciplinary laboratory on emergencies such as pandemics, dealing with research and development, on a regional/national-

wide scale thus able to offer consultancy and to even develop resilient solutions to such kind of problems.

From a more technical point of view, we are investigating new techniques and approaches to perform more accurate and reliable clustering and automate the taxonomy generation process to further revise and extend the Covid19 one, also including other disciplines and application contexts.

## Author Contributions

**Conceptualization:** Salvatore Distefano.

**Data curation:** Giovanni Cicceri, Letterio Galletta, Marco Polignano, Carlo Scaffidi.

**Formal analysis:** Vincenzo Bonnici, Giovanni Cicceri, Salvatore Distefano, Marco Polignano.

**Funding acquisition:** Salvatore Distefano.

**Investigation:** Vincenzo Bonnici, Salvatore Distefano, Letterio Galletta, Marco Polignano, Carlo Scaffidi.

**Methodology:** Vincenzo Bonnici, Giovanni Cicceri, Salvatore Distefano, Letterio Galletta, Marco Polignano, Carlo Scaffidi.

**Project administration:** Salvatore Distefano.

**Software:** Giovanni Cicceri, Letterio Galletta, Marco Polignano.

**Supervision:** Salvatore Distefano.

**Validation:** Salvatore Distefano.

**Visualization:** Vincenzo Bonnici, Giovanni Cicceri, Salvatore Distefano, Letterio Galletta, Marco Polignano.

**Writing – original draft:** Vincenzo Bonnici, Giovanni Cicceri, Salvatore Distefano, Letterio Galletta, Marco Polignano, Carlo Scaffidi.

**Writing – review & editing:** Vincenzo Bonnici, Giovanni Cicceri, Salvatore Distefano, Letterio Galletta, Marco Polignano, Carlo Scaffidi.

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
