## [Decision Letter · Decision Letter 0]

11 Apr 2022

PONE-D-22-03195Covid19/IT - the digital side of Covid19: a picture from Italy with clustering and taxonomyPLOS ONE

Dear Dr. Distefano,

Thank you for submitting your manuscript to PLOS ONE. After careful consideration, we feel that it has merit but does not fully meet PLOS ONE’s publication criteria as it currently stands. Therefore, we invite you to submit a revised version of the manuscript that addresses the points raised during the review process.

We look forward to receiving your revised manuscript.

Kind regards,

Chi-Hua Chen, Ph.D.

Academic Editor

PLOS ONE

Journal Requirements:

Reviewers' comments:

Reviewer's Responses to Questions

**Comments to the Author**

1. Is the manuscript technically sound, and do the data support the conclusions?

Reviewer #1: Partly

2. Has the statistical analysis been performed appropriately and rigorously? 

Reviewer #1: No

3. Have the authors made all data underlying the findings in their manuscript fully available?

Reviewer #1: Yes

4. Is the manuscript presented in an intelligible fashion and written in standard English?

Reviewer #1: No

5. Review Comments to the Author

Reviewer #1: Introduction and Literature Review should be split in two different sections.

The Introduction should highlight the relevance of the topic, the novelty of the results, the importance of policy implications, the sample’s choice, the methodology’s appropriateness, the data used, the contribution to the literature, and the limitations of the study.

The literature review is partial and incomplete, and some recent and relevant contributions should be cited and discussed: i.e., 10.1017/S095026882100248X; 10.1016/j.jenvman.2021.112241; 10.1007/s11356-020-10689-0; 10.1016/j.envres.2020.110663; 10.1016/j.apenergy.2020.115835.

The theoretical framework should be discussed more in detail.

The estimated model must be justified in light of the literature on this specific topic.

Descriptive statistics are absent.

Diagnostic tests are absent.

Robustness checks are absent.

The results should be discussed more in detail.

Comparisons with previous studies are absent.

Conclusions are too short.

Policy implications are weak.

Further research should be indicated.

Limitations of the study are not provided.

The English needs a proofreading by a native speaker.

The editing does not follow the journal’s guidelines.

The originality value of the study is limited.

6. PLOS authors have the option to publish the peer review history of their article (what does this mean?). If published, this will include your full peer review and any attached files.

Reviewer #1: No

---

## [Author Response · Author response to Decision Letter 0]

23 May 2022

Reviewer #1: 

Q1:Introduction and Literature Review should be split in two different sections.

The Introduction should highlight the relevance of the topic, the novelty of the results, the importance of policy implications, the sample’s choice, the methodology’s appropriateness, the data used, the contribution to the literature, and the limitations of the study.

A1: The Introduction and literature review have been split into two different sections, then further revised and extended. Specifically, the introduction has been reorganized and rewritten to address this comment, with a specific subsection for each of the suggested parts that are missing in the original version of the paper. Thereby, 6 subsections have been added to the introduction: Relevance of the topic, Novelty of the results, Research impact and policy implications, Methodology, theoretical framework and limitations of the study, Contribution to the literature, and Outline.

Q2: The literature review is partial and incomplete, and some recent and relevant contributions should be cited and discussed: i.e., 10.1017/S095026882100248X; 10.1016/j.jenvman.2021.112241; 10.1007/s11356-020-10689-0; 10.1016/j.envres.2020.110663; 10.1016/j.apenergy.2020.115835.

A2: We revised our literature review taking into account the references suggested by this Reviewer and adding new ones. We cited them in the sub-section entitled “Covid19 Datasets, Taxonomies and Ontologies” highlighting the main difference between the dataset proposed in those papers and our dataset. More Precisely, we remarked that the goal of our dataset is to collect about the research projects against Covid19 on the Italian territory, in all topics related to computer science, engineering and technologies. Whereas the previous work in the literature proposed datasets about specific aspects related to Covid-19. 

Q3: The theoretical framework should be discussed more in detail.

A3: A theoretical framework discussion was missing in the original version of the paper. To fill this gap, in the revised version of the paper we added a specific subsection in the introduction, reporting and summarizing the overall paper theoretical framework, and ad-hoc paragraphs in the Clustering and Taxonomy sections, with further and more specific details.

Q4: The estimated model must be justified in light of the literature on this specific topic.

A4: The models adopted in the paper, mainly in the clustering and taxonomy steps, have been commented on, adding new descriptions in the corresponding sections. More specifically, in the Clustering section, the word embedding, NLP and k-means models, as well as t-SNE and PCA ones, have been further explained, detailed and compared to literature. In the Taxonomy section, details on the original taxonomy model have been added.

Q5: Descriptive statistics are absent.

A5: We thank the Reviewer for this comment that helped us to better describe the section “Data Analysis”. The session has been renamed to “Descriptive Data Analysis” and revised accordingly to better describe the descriptive statistics that have been performed in the work. We also revised the section to better emphasize aims and goals of the analyses.

Q6: Diagnostic tests are absent. Robustness checks are absent.

A6: Diagnostic and robustness tests were missing in the original version of the paper and have been performed and described in the revised version. A description has been added regarding the reliability and robustness analysis of the clusters obtained through the NLP approach proposed in the paper. The metrics of Adjusted Rand Index, Normalized Mutual-Information, Homogeneity, Completeness, V1 score, Fowlkes-Mallows index have been calculated and discussed in the subsection "Experiments and Results".

Q7: The results should be discussed more in detail.

A7: A more detailed qualitative analysis regarding the obtained clusters has been added to the subsection "Cluster visualization and interpretability".

Q8: Comparisons with previous studies are absent.

A8: In accordance with the Reviewer comment, we entirely revised the literature review - Related Work - section. More specifically, at the end of each sub-section a paragraph has been added to point out the main novelties and differences between the solution proposed by this paper and the existing ones before referenced by the literature. Furthermore, specific paragraphs have been added in the Clustering and Taxonomy sections to compare the specific approach they adopted against existing ones.

Q9: Conclusions are too short.

Policy implications are weak.

Further research should be indicated.

A9: We extended the final section about the lesson learned, conclusions and future work with new discussions and considerations. Specifically, potential policy implications have been identified, described and commented, both in the introduction and (mainly) in the latest section. Similarly, future work describing the new activities we plan in the future have been detailed. In particular, we described our ideas of carrying out a new survey to observe how the previously surveyed activities have been completed. Furthermore we discussed the creation of a multi-disciplinary laboratory about emergencies such as pandemics. This laboratory will support Covid19/pandemic-related research and development on a regional/national-wide scope.

Q10: Limitations of the study are not provided.

A10: We thank the Reviewer for pointing out this missing aspect. We identified and added the limitations of our work in the Introduction as a subsection “Methodology, theoretical framework and limitations of the study” and further detailed them in the section “Lesson learned, conclusions and future work”. 

Q11: The English needs a proofreading by a native speaker.

A11: The paper has been revised thoroughly to remove typos and misleading statements. An in depth proofreading check has been performed by our university linguistic centers.

Q12: The editing does not follow the journal’s guidelines.

A12: We thank the Reviewer for this check. We revised the manuscript to comply with the journal guidelines. In particular, we replaced footnotes with references, and we placed all the figures and tables immediately after the first paragraph they are cited.

Q13: The originality value of the study is limited.

A13: We revised the paper to better highlight the contribution of the work, in particular the Introduction, by adding new specific subsections that better present the relevance, the originality and the impact of our work. Moreover, the novelty of the paper has been further detailed in the technical sections, i.e. Related work, Clustering and Taxonomy ones, in comparison with existing solutions to remark the contribution.

---

## [Decision Letter · Decision Letter 1]

26 May 2022

Covid19/IT - the digital side of Covid19: a picture from Italy with clustering and taxonomy

PONE-D-22-03195R1

Dear Dr. Distefano,

We’re pleased to inform you that your manuscript has been judged scientifically suitable for publication and will be formally accepted for publication once it meets all outstanding technical requirements.

Kind regards,

Chi-Hua Chen, Ph.D.

Academic Editor

PLOS ONE

Additional Editor Comments (optional):

Reviewers' comments:

Reviewer's Responses to Questions

**Comments to the Author**

1. If the authors have adequately addressed your comments raised in a previous round of review and you feel that this manuscript is now acceptable for publication, you may indicate that here to bypass the “Comments to the Author” section, enter your conflict of interest statement in the “Confidential to Editor” section, and submit your "Accept" recommendation.

Reviewer #1: All comments have been addressed

2. Is the manuscript technically sound, and do the data support the conclusions?

Reviewer #1: Yes

3. Has the statistical analysis been performed appropriately and rigorously? 

Reviewer #1: Yes

4. Have the authors made all data underlying the findings in their manuscript fully available?

Reviewer #1: Yes

5. Is the manuscript presented in an intelligible fashion and written in standard English?

Reviewer #1: Yes

6. Review Comments to the Author

Reviewer #1: Accept ----------------------------------------------------------------------------------------------------------

7. PLOS authors have the option to publish the peer review history of their article (what does this mean?). If published, this will include your full peer review and any attached files.

Reviewer #1: No

---

## [Editor Report · Acceptance letter]

31 May 2022

PONE-D-22-03195R1 

Covid19/IT the digital side of Covid19: a picture from Italy with clustering and taxonomy

Dear Dr. Distefano:

I'm pleased to inform you that your manuscript has been deemed suitable for publication in PLOS ONE. Congratulations! Your manuscript is now with our production department. 

Kind regards, 

on behalf of

Professor Chi-Hua Chen 

Academic Editor

PLOS ONE